# ParaSolver: A Hierarchical Parallel Integral Solver for Diffusion Models

**Jianrong Lu, Zhiyu Zhu, and Junhui Hou**[*]
[1]Department of Computer Science, City University of Hong Kong, Hong Kong SAR
`jrong.alvin@gmail.com, zhiyuzhu2-c@my.cityu.edu.hk,`
`jh.hou@cityu.edu.hk`

## Abstract

This paper explores the challenge of accelerating the sequential inference process of Diffusion Probabilistic Models (DPMs). We tackle this critical issue from a dynamic systems perspective, in which the inherent sequential nature is transformed into a parallel sampling process. Specifically, we propose a unified framework that generalizes the sequential sampling process of DPMs as solving a system of banded nonlinear equations. Under this generic framework, we reveal that the Jacobian of the banded nonlinear equations system possesses a unit-diagonal structure, enabling further approximation for acceleration. Moreover, we theoretically propose an effective initialization approach for parallel sampling methods. Finally, we construct *ParaSolver*, a hierarchical parallel sampling technique that enhances sampling speed without compromising quality. Extensive experiments show that ParaSolver achieves up to **12.1× speedup** in terms of wall-clock time. The source code is publicly available at
https://github.com/Jianrong-Lu/ParaSolver.git.

## 1 Introduction

Over the past few years, the landscape of generative modeling has been significantly reshaped by the ascent of diffusion probabilistic models (DPMs) (Ho et al., 2020; Song et al., b). These models have emerged as a pivotal methodology for diverse applications (Yang et al., 2024; Liu et al., 2023a; Lu et al., 2024; Chung et al., 2023; Lu et al., 2023; You et al., 2025; Liu et al., 2025), spanning from high-quality image/video synthesis (Rombach et al., 2022; Blattmann et al., 2023) to molecular generation (Wu et al., 2024; Nguyen et al., 2023). In essence, diffusion models hinge on a noise reduction process that is represented mathematically as an ordinary or stochastic differential equation (ODE/SDE). The equation systematically removes noise from an initial normal distribution, transforming it into a vivid sample that conforms to the intended real data distribution, thereby facilitating high-quality generative modeling. To produce high-quality samples, nevertheless, DPMs typically necessitate a multitude of sequential noise reduction iterations involving extensive evaluations of large neural networks, leading to a notably slow sampling speed.

Researchers have put forth a range of methods (Gong et al., 2024; Zheng et al., 2023; Gonzalez et al., 2024; Geng et al., 2024; Liu et al., 2023b; Luo et al., 2023) to improve sampling speed. One strategy involves developing faster SDE/ODE solvers like DDIM (Song et al., a) and DPMSolver (Lu et al., 2022) based on mathematical principles to expedite the sampling process. However, these approaches often require a reduction in the number of denoising steps, which impacts the quality of the samples. Another category is to distill the ODE trajectory of pre-trained diffusion model into another neural network that enables short-step sampling, exemplified by work such as (Salimans & Ho; Song et al., 2023). An additional enhancement for this kind of method is to straighten the convoluted ODE trajectory of DPMs into a straight line using the Rectified Flow (Lipman et al.; Liu et al.). Nevertheless, it is common for these approaches to result in a decline in both image quality and diversity.

---

[*]The first two authors contributed to this work equally. Corresponding authors: Zhiyu Zhu and Junhui Hou. This project was supported in part by the NSFC Excellent Young Scientists Fund 62422118, and in part by the Hong Kong RGC under Grants 11219422, 11219324 and 11218121.

To circumvent the aforementioned questions and expedite the sampling process, Shih et al. (2024b) introduced a parallelization technique that simultaneously denoises multiple steps through Picard iteration. Subsequently, Tang et al. (2024) further redefined the parallel sampling paradigm of diffusion models, framing it as the solution of a series of triangular nonlinear equations via fixed-point iteration (FP). These methodologies present a trio of distinct benefits compared to existing approaches: (**1**) training-free and compatible with existing fast sampling methods; (**2**) yielding samples with comparable quality to sequential sampling; (**3**) resulting in a notable decrease in sampling steps, thereby greatly enhancing sampling efficiency.

To further extend the capabilities of the parallel methods and enable new sampling paradigms, we propose *ParaSolver*, a unified framework that generalizes previous approaches through the lens of nonlinear equations (NEs). Specifically, we formulate the sampling process of DPMs as a system of NEs with a computation-efficient banded structure. This framework leads to parallel sampling in a hierarchical way, thus improving the efficiency of parallel sampling methods in situations with constrained computing resources. Moreover, any current sequential or parallel sampling algorithms for DPMs can be seamlessly integrated into our framework.

In summary, this paper encompasses the following theoretical and practical contributions:

- we propose a novel parallel sampling algorithm for diffusion models named ParaSolver, via partitioning the sequential inference process into a system of non-linear equations;

- we accelerate and simplify the updating process via the unique structure of the Jacobian matrix of the system, which has a unit-diagonal structure with compute-intensive gradients in the sub-diagonal;

- we conduct extensive experiments, showcasing that our ParaSolver can achieve $2\times \sim 12\times$ **speedup** in terms of wall-clock time, establishing a new record in this field.

## 2 RELATED WORK

Parallel sampling that enables faster sampling without quality degradation opens up a promising avenue for faster sampling of DPMs. It concurrently denoises multiple sample points on the ODE/SDE trajectories of DPMs by iteratively refining them from an array of initial guesses. To the best of our knowledge, the recent studies Shih et al. (2024a) and Tang et al. (2024) are the only research endeavors concentrating on parallel sampling methods for DPMs. ParaDiGMS (Shih et al., 2024a) represents the pioneering parallel sampling approach. It treats the discretized sample points along the ODE trajectories as a sequence of fixed points. These points are progressively enhanced through iterations using the established fixed-point theorem and Picard-Lindelöf theorem. ParaTAA (Tang et al., 2024) extends the fixed-point iteration approach by defining the sampling process of DPMs as solving triangular nonlinear equations, with all discretized sample points acting as unknown variables. In contrast to ParaTAA, we frame the sampling procedure of DPMs as a set of banded nonlinear equations. This leads to a hierarchical parallel method, which is totally different from existing methods parallelizing all points in the ODE trajectories. Moreover, this banded structure, characterized by its sparsity, facilitates significantly more efficient computations compared to the compute-intensive triangular structure. Additionally, ParaTAA can be seamlessly incorporated into our framework. We have also identified intriguing research areas dedicated to expediting diffusion models. These include topics like model or sample partitioning (Ma et al., 2024b;a; Wang et al., 2024; Li et al., 2024), trajectory stitching (Pan et al., 2024), nested diffusion (Elata et al., 2024), minimum denoising step prediction (Yu & Barad, 2024), and theoretical assurances (Gupta et al., 2024; Chen et al., 2024). Our strategy revolves around parallelizing the sequential sampling process of DPMs, which is complementary to theirs.

While these methods show promising benefits, several drawbacks constrain their parallel efficiency. First, parallelizing all discretized sample points simultaneously becomes compute-intensive when handling large models, leading to a rapid degradation in parallel efficiency. Second, the final clean sample relies on all samples from previous timesteps. As a result, errors from these prior sample points all accumulate in the final clean samples. This, in turn, necessitates additional iterations to correct these accumulated errors, further restricting parallel efficiency.

## 3 PRELIMINARY

**Diffusion Models**. Denote by $\mathbf{X}_t \in \mathbb{R}^{HWC}$ the noised variable on the diffusion trajectory, $t \in [0, T]$ the scalar indicating the time-stamp, $f(t)$ the drift coefficient function, and $g(t)$ the diffusion coefficient function. Then, the forward diffusion process can be formulated as the following stochastic differential equation (SDE), which projects the clean image $\mathbf{X}_T$ to a random noise $\mathbf{X}_0$ (we define the clean image at time $T$ and the noise at time 0),

$$d\mathbf{X}_t = f(t)\mathbf{X}_t dt + g(t)d\mathbf{W}, \tag{1}$$

where $d\mathbf{W}$ indicates the standard Wiener process. Then, to generate the corresponding clean latent from the easily sampled random noise, we have to reverse the forward SDE in Eq. (1), resulting in the following reverse SDE formulations

$$d\mathbf{X}_t = \underbrace{\left[f(t)\mathbf{X}_t - g^2(t)\nabla_{\mathbf{X}_t} \log p(\mathbf{X}_t)\right]}_{\varphi(\mathbf{X}_t, t)} dt + \underbrace{g(t)}_{\sigma_t} d\mathbf{W}, \tag{2}$$

where $\nabla_{\mathbf{X}_t} \log p(\mathbf{X}_t)$ can be approximated by a score function $\mathbf{S}_\theta(\cdot)$, parameterized by a neural network with learnable weights of $\theta$; $\varphi(\mathbf{X}_t, t)$ denotes the drift function for the reverse diffusion process; $\sigma_t$ represents the corresponding coefficient of diffusion counterpart. Let $\Phi(t, s, \mathbf{X}_s)$ represent an integral result of $\mathbf{X}_t$ by Eq. (2) over a time interval from $s$ to $t$, with an initial value $\mathbf{X}_s$:

$$\Phi(t, s, \mathbf{X}_s) = \mathbf{X}_s + \int_s^t \varphi(\mathbf{X}_\tau, \tau)d\tau + \int_s^t \sigma_\tau d\mathbf{W}. \tag{3}$$

Consequently, the analytical solution of Eq. (2) at time $t$ can be expressed as

$$\mathbf{X}_t = \Phi(t, 0, \mathbf{X}_0), \mathbf{X}_0 \sim \mathcal{N}(\mathbf{0}, \boldsymbol{I}), \tag{4}$$

where $\mathcal{N}(\mathbf{0}, \boldsymbol{I})$ denotes the standard Gaussian distribution.

**Parallel Sampling Algorithms**. For an array of ODE/SDE trajectories for DPMs $\{\mathbf{X}_t, t = 0, \cdots, T\}$[1], existing parallel sampling algorithms (Shih et al., 2024a; Tang et al., 2024) establish the following system of non-linear equations to reformulate the integral-based formulation of the diffusion model:

$$\mathbf{X}_{t+1} - \mathcal{H}_t^{(i)}(\mathbf{X}_t, \cdots, \mathbf{X}_{t-i}) = \mathbf{0}, \tag{5}$$

where $i$ is the number of utilizing previous states at timestamp $t$; $\mathcal{H}_t^{(i)}$ denotes a solver for estimating results in timestamp $t$ with acknowledging previous states, i.e., $\mathbf{X}_t, \cdots, \mathbf{X}_{t-i}$.

The sampling methods utilize an iterative refinement manner to gradually adjust an estimation trajectory $\left\{\hat{\mathbf{X}}_t, t = 0, \cdots, T\right\}$. Each state from the trajectory $\{\mathbf{X}_t, t = 0, \cdots, T\}$ is first initialized with coarse, even noise value, denoted as $\left\{\hat{\mathbf{X}}_t^{(0)}, t = 0, \cdots, T\right\}$. Denote by $\hat{\mathbf{X}}_t$ the vector, $\hat{\mathbf{X}}_{0:T} = [\hat{\mathbf{X}}_0^\top, \cdots, \hat{\mathbf{X}}_T^\top,]^\top$. Then, for the $k$-th parallel iteration ($k \in [0, K]$), Newton–Raphson method (Kelley, 2003) updates the variables by the following scheme

$$\hat{\mathbf{X}}_{0:T}^{(k+1)} = \hat{\mathbf{X}}_{0:T}^{(k)} - \mathcal{G}^k \mathcal{Q}_{0:T}^{(k)}, \tag{6}$$

where $\mathcal{Q}_t^{(k)} = \hat{\mathbf{X}}_t^{(k)} - \mathcal{H}_t^{(i)}(\hat{\mathbf{X}}_t^{(k)}, \cdots, \hat{\mathbf{X}}_{t-i}^{(k)})$ indicates a residual term to be optimized; and $\mathcal{G}^k = (\mathcal{J}^{(k)})^{-1}$ indicates the inverse of Jacobian matrix $\mathcal{J}^{(k)} = \frac{\partial \mathcal{Q}_{0:T}^{(k)}}{\partial \hat{\mathbf{X}}_{0:T}}$.

## 4 PROPOSED METHOD

Parallel solvers transform the sequential integral process of the diffusion model, as represented in Eq. (4), into the view of non-linear equations articulated in Eq. (6). However, the typically large number of timestamps complicates the simultaneous solving of all nodes, particularly with a limited number of GPU/CPU cores. To mitigate this issue, we propose a hybrid approach that integrates the fully sequential inference of classical diffusion models with comprehensive parallel methodologies. Specifically, we reformulate the partitioning of non-linear equations using a banded structure, which leverages the strengths of both parallel and sequential techniques, as discussed in Sec. 4.1. Furthermore, based on this formulation, we analyze strategies to simplify and accelerate the solving process with corresponding steps for initialization and stopping criterion in Sec. 4.2.

---

[1]To adapt the parallel sampling algorithm, we discretize the continuous trajectory $[0, T]$ into several intervals $[0, 1, \cdots, T]$ by a step size of 1.

## 4.1 Hierarchical Sampling via Banded Non-linear Equations

We start with the re-investigation of the integral formulation of diffusion process Eq. (4) in the following partitioned manner,

$$\mathbf{X}_{t_{n+1}} = \Phi(t_{n+1}, t_n, \mathbf{X}_{t_n}), n \in \{0, 1, \cdots, N-1\}, \tag{7}$$

where the whole temporal region of $[0, T]$ is partitioned into $N$ time sub-intervals, with $1 \leqslant N \leqslant T$ and $0 = t_0 < t_1 < \cdots t_N = T$[2]. Inspired by current parallel algorithms, e.g., Song et al. (2021), such a sequential problem can be solved in a parallel manner. Specifically, such a series of cascaded functions can be treated as a system of NEs with a banded structure. Denote by $\{\mathbf{X}_{t_n}, n = 0, \cdots, N\}$ a set of points exactly *on* the diffusion trajectory, $\left\{\hat{\mathbf{X}}_{t_n}, n = 0, \cdots, N\right\}$ the points to be optimized approaching the trajectory. A cascade of local integral formulations as Eq. (7) can be reformulated as the solutions of following *Banded NEs*.

**Definition 1** (Banded NEs). *We define the system of banded NEs for the sequential sampling process in* Eq. (7) *as*

$$\begin{cases} \hat{\mathbf{X}}_{t_0} - \mathbf{X}_{t_0} = \mathbf{0}, \\ \hat{\mathbf{X}}_{t_{n+1}} - \Phi(t_{n+1}, t_n, \hat{\mathbf{X}}_{t_n}) = \mathbf{0}, n \in \{0, 1, \cdots, N-1\}. \end{cases} \tag{8}$$

Moreover, we also have a theoretical analysis that the solution of Eq. (8) is the unique and unbiased estimation of the sequential sampling results, as shown in Prop. 1.

**Proposition 1** (Unbiased Estimation). *The set of banded NEs in* Eq. (8) *possesses a unique solution, which is also an unbiased estimation of the sequential diffusion results.*

*Proof.* See Appendix A. □

By the aforementioned, we reformulate the sequential sampling process of the diffusion model into two hierarchical phases, i.e., globally solving the banded NEs of Eq. (8) and locally calculating the integral process $\Phi(\cdot)$. Each of them can be solved parallel or sequential by existing methods, which distinguishes the proposed method compared to other sequential or parallel sampling methods. Since solving the global non-linear equations is the primary bottleneck for the acceleration of diffusion sampling, we will mainly discuss the methods to further simplify and accelerate solving Eq. (8).

*Novelty Claim*. Due to the aforementioned hierarchical sampling manner, the system of NEs in our proposed method has smaller-scale NEs and variables than the existing parallel sampling algorithms and features a notably sparse banded structure. Consequently, hierarchical sampling can conserve numerous computational resources than the existing methods with a dense triangular structure. Moreover, we want to note that our framework encapsulates the existing parallel and sequential methods. The banded NEs with $N = 1$ and $N = T$ are precisely aligned with the sequential sampling method in Eq. (4) and the parallel sampling approach in Eq. (5), respectively.

## 4.2 Solving the System of Banded NEs

We investigate the Newton–Raphson method (Kelley, 2003) to solve the aforementioned NEs as Eq. (8), as

$$\hat{\mathbf{X}}_{t_0:t_N}^{(k+1)} = \hat{\mathbf{X}}_{t_0:t_N}^{(k)} - \left(\mathcal{J}_{t_0:t_N}^{(k)}\right)^{-1} \mathcal{R}_{t_0:t_N}^{(k)}, \tag{9}$$

where $\mathcal{R}_{t_n}^{(k)} = \hat{\mathbf{X}}_{t_{n+1}}^{(k)} - \Phi(t_{n+1}, t_n, \hat{\mathbf{X}}_{t_n}^{(k)})$ indicates the residual term based on our previously formulated NEs; $\left(\mathcal{J}_{t_0:t_N}^{(k)}\right)^{-1}$ represents the inverse of Jacobian matrix as $\partial \mathcal{R}_{t_0:t_N}^{(k)} / \partial \hat{\mathbf{X}}_{t_0:t_N}^{(k)}$.

**Proposition 2** (Smoothness Analysis). $\Phi(t_{n+1}, t_n, \hat{\mathbf{X}}_{t_n})$ *satisfies* $\left\| \frac{\partial \Phi(t_{n+1}, t_n, \hat{\mathbf{X}}_{t_n})}{\partial \hat{\mathbf{X}}_{t_n}} \right\|_F \leqslant \frac{\alpha_{t_{n+1}}}{\alpha_{t_n}} + \gamma \sigma_{t_{n+1}} \left( \frac{\alpha_{t_{n+1}} \sigma_{t_n}}{\alpha_{t_n} \sigma_{t_{n+1}}} - 1 \right) \approx 1$, *where* $0 \leqslant \gamma < 1$; $\alpha$ *and* $\sigma$ *are the noise schedule for the forward process* $\mathcal{N}(\alpha_{t_n} \mathbf{X}_{t_N}, \sigma_{t_n} \mathbf{I})$ *from clean image* $\mathbf{X}_{t_N}$ *to noisy image at* $t_n$. *This small upper bound leads to sufficiently smooth* $\Phi$, *implying* $\mathcal{R}_{t_n}$ *is smooth enough.*

---

[2]We discretize the continuous trajectory $[0, T]$ into several intervals $[t_0, t_1, \cdots, t_N]$ by a step size of not less than 1.

*Proof.* See Appendix B. □

***Applicability analysis***. Proposition 2 suggests that the residual for the nonlinear system in Definition 1 has a sufficiently smooth landscape, enabling Newton's method for root-finding to converge rapidly without getting stuck easily.

Although, through such a manner, we seem to be solving the problem efficiently in a parallel manner, the calculation of the reverse Jacobian matrix is computationally complex, even unsolvable.

Inspired by the classical optimization theory in NEs (Wolfe, 1959; Broyden, 1965; Anderson, 1965), we approximate the reverse Jacobian of the NEs system and execute a Jacobian-like update based on this estimation. Our initial crucial insight is that we can precisely calculate the diagonal component of the Jacobian by taking into account the specific structure of the banded NEs system. The residual term of the banded NEs system comprises two components: the part $\hat{\mathbf{X}}_{t_0:t_N}^{(k)}$ behaves as linear functions of itself, while the remaining part of $\Phi\left(t_1 : t_N, t_0 : t_{N-1}, \hat{\mathbf{X}}_{t_0:t_N}^{(k)}\right)$ generally represents nonlinear functions of $\hat{\mathbf{X}}_{t_0:t_N}^{(k)}$ because it involves the neural network $\boldsymbol{S}_\theta\left(\hat{\mathbf{X}}_{t_0:t_N}^{(k)}\right)$. These semi-linear NEs are ignorant by existing method (Tang et al., 2024) as it estimates the entire triangular part of the Jacobian matrix, which causes estimated errors in the linear part.

We note that for the semi-linear banded NEs system, the reverse Jacobian at $k^{th}$ parallel iteration can be exactly formulated as a unit lower block banded matrix:

$$\mathbf{B}_{i,j} = \begin{cases} \mathbf{I}, & \text{if } j = i \\ \frac{\partial \Phi\left(t_i, t_j, \hat{\mathbf{X}}_{t_j}^{(k)}\right)}{\partial \hat{\mathbf{X}}_{t_j}^{(k)}}, & \text{if } j = i - 1 \\ \mathbf{0}, & \text{otherwise} \end{cases} \tag{10}$$

where $\mathbf{B}_{i,j}$ is a block matrix within the unit lower block banded matrix $\left(\mathcal{J}_{t_0:t_N}^{(k)}\right)^{-1}$ at row $i$ and column $j$. Based on this insight, we can obtain an exact parallel sampling iteration for our banded NEs system.

**Proposition 3** (Exact Parallel Sampling). *Given a set of initial values $\{\hat{\mathbf{X}}_{t_n}^{(0)}, n = 0, 1, \cdots, N\}$, the exact parallel recurrence for our semi-linear banded NEs system in Definition 1 at parallel iteration $k$ is*

$$\begin{cases} \hat{\mathbf{X}}_{t_0}^{(k+1)} = \hat{\mathbf{X}}_{t_0}^{(0)} = \mathbf{X}_{t_0} \sim \mathcal{N}(\mathbf{0}, \mathbf{I}), \\ \hat{\mathbf{X}}_{t_{n+1}}^{(k+1)} = \Phi(t_{n+1}, t_n, \hat{\mathbf{X}}_{t_n}^{(k)}) + \frac{\partial \Phi(t_{n+1}, t_n, \hat{\mathbf{X}}_{t_n}^{(k)})}{\partial \hat{\mathbf{X}}_{t_n}^{(k)}} \left(\hat{\mathbf{X}}_{t_n}^{(k+1)} - \hat{\mathbf{X}}_{t_n}^{(k)}\right). \end{cases} \tag{11}$$

*Proof.* See Appendix C. □

For the nonlinear part, i.e., the Jacobian term $\frac{\partial \Phi(t_{n+1}, t_n, \hat{\mathbf{X}}_{t_n}^{(k)})}{\partial \hat{\mathbf{X}}_{t_n}^{(k)}}$, in practice, is expensive to compute. Following numerous studies on Jacobian-free backpropagation (Fung et al., 2022; Geng et al.; Liu et al., 2024; Knoll & Keyes, 2004; Poole et al.), we find that substituting the Jacobian with the identity matrix cannot impact the iteration process. This yields the following effective recurrence:

$$\begin{cases} \hat{\mathbf{X}}_{t_0}^{(k+1)} = \hat{\mathbf{X}}_{t_0}^{(0)} = \mathbf{X}_{t_0} \sim \mathcal{N}(\mathbf{0}, \mathbf{I}), \\ \hat{\mathbf{X}}_{t_{n+1}}^{(k+1)} = \Phi(t_{n+1}, t_n, \hat{\mathbf{X}}_{t_n}^{(k)}) + \hat{\mathbf{X}}_{t_n}^{(k+1)} - \hat{\mathbf{X}}_{t_n}^{(k)}. \end{cases} \tag{12}$$

Reorganizing Eq. (12) leads to the formula of its general term

$$\begin{cases} \hat{\mathbf{X}}_{t_0}^{(k+1)} = \hat{\mathbf{X}}_{t_0}^{(0)} = \mathbf{X}_{t_0} \sim \mathcal{N}(\mathbf{0}, \mathbf{I}), \\ \hat{\mathbf{X}}_{t_{n+1}}^{(k+1)} = \hat{\mathbf{X}}_{t_0}^{(k+1)} + \sum_{i=0}^{n} \Phi(t_{i+1}, t_i, \hat{\mathbf{X}}_{t_i}^{(k)}) - \sum_{i=0}^{n} \hat{\mathbf{X}}_{t_i}^{(k)}. \end{cases} \tag{13}$$

**Proposition 4** (Convergence Analysis). *The update rule in Eq. (13) has a descent direction that is not contradictory to the exact update rule in Eq. (11) for the residual $\mathcal{R}_{t_0:t_N}^{(k)}$. Its convergence speed falls between linear and quadratic convergence.*

*Proof.* See Appendix D. □

**Stopping Criterion.** We need to establish a stopping criterion for our parallel sampling to prevent any decline in sample quality. Setting a sufficiently low tolerance ensures that the outcome of parallel sampling closely resembles that of the sequential sampling procedure. Specifically, denote by $\delta$ the tolerance. We follow Shih et al. (2024a) to define the stopping criterion as

$$\frac{1}{D}\left\|\hat{\mathbf{X}}_{t_n}^{(k+1)} - \hat{\mathbf{X}}_{t_n}^{(k)}\right\|_F^2 \leqslant \delta^2 \sigma_{t_n}^2, \tag{14}$$

where $D$ is the dimensional of sample $\hat{\mathbf{X}}_{t_n}$; $\|\cdot\|_F$ is Frobenius norm. As demonstrated in Shih et al. (2024a), it has been established that this relaxed stopping criterion can guarantee that the samples of $\boldsymbol{X}_{t_N}^K$ are obtained from a distribution that exhibits a minimal total variation distance from the DDPM model distribution.

**Initialization.** The parallel iteration in Eq. (13) commences with an array of initial values $\{\hat{\mathbf{X}}_{t_n}^{(0)}, n = 0, \cdots, N\}$. Initiating the iterations with values that closely approximate the solutions $\{\mathbf{X}_{t_n}, n = 0, \cdots, N\}$ will lead to swift convergence. Hence, initialization also stands as a crucial component for ensuring parallel efficiency. To review the banded NEs system in Definition 1, we find that the initializations should satisfy the condition $\hat{\mathbf{X}}_{t_{n+1}}^{(0)} = \Phi(t_{n+1}, t_n, \hat{\mathbf{X}}_{t_n}^{(0)})$ with $\hat{\mathbf{X}}_{t_0}^{(0)} \sim \mathcal{N}(\mathbf{0}, \mathbf{I})$. However, executing such a sampling procedure, whether in parallel or sequentially across $N + 1$ time points $t_0, \cdots, t_N$, which may be large, usually requires a significant amount of time.

Given this, our core insight is to utilize the principle of DPMs to fast construct a set of more precise initial values that conform to the Definition 1. In particular, the condition $\mathbf{X}_{t_{n+1}} = \Phi(t_{n+1}, t_n, \mathbf{X}_{t_n})$ can be estimated by the reverse process using coarse timesteps from $t_n$ to $t_{n+1}$:

$$\mathbf{X}_{t_{n+1}} \sim q(\mathbf{X}_{t_{n+1}} | \mathbf{X}_{t_n}, \mathbf{X}_{t_N}), \tag{15}$$

where $q$ is the transition probability distribution from $\mathbf{X}_{t_n}$ to $\mathbf{X}_{t_{n+1}}$ for the reverse process, which becomes traceable when conditioned on the clean sample $\mathbf{X}_{t_N}$ (we define the clean image at time $t_N = T$ and the noise at time $t_0 = 0$). According to the forward process $\mathbf{X}_{t_n} \sim \mathcal{N}(\alpha_{t_n} \mathbf{X}_{t_N}, \sigma_{t_n} \mathbf{I})$ where $\alpha$ and $\sigma$ are the noise schedule, the clean sample at parallel iteration $k$ can be estimated by the neural network $\boldsymbol{S}_\theta(\mathbf{X}_{t_n}, t_n)$ at time $t_n$:

$$\hat{\mathbf{X}}_{t_N}^{(k)}(t_n) = \frac{\mathbf{X}_{t_n}^{(k)} - \sigma_{t_n} \boldsymbol{S}_\theta(\mathbf{X}_{t_n}^{(k)}, t_n)}{\alpha_{t_n}}. \tag{16}$$

Once obtaining the estimated clean sample $\hat{\mathbf{X}}_{t_N}$, we can then approximate all the remaining noisy samples via the reverse process at negligible costs. To initialize the diffusion trajectory, we can perform a few steps of the reverse process as preconditioning steps, with minimal expense. Denote by $M$ the number of the preconditioning steps. Then, after $M$ steps of the reverse process, we can approximate the clean sample by Eq. (16) using $\hat{\mathbf{X}}_{t_N}^{(0)}(t_{M-1})$. Formally, we define the set of initialized values as follows:

$$\begin{cases} \hat{\mathbf{X}}_{t_n}^{(0)} = \mathbf{X}_{t_n}, & \text{if } 0 \leqslant n < M \\ \hat{\mathbf{X}}_{t_{n+1}}^{(0)} \sim q(\hat{\mathbf{X}}_{t_{n+1}}^{(0)} | \hat{\mathbf{X}}_{t_n}^{(0)}, \hat{\mathbf{X}}_{t_N}^{(0)}(t_{M-1})), & \text{if } M \leqslant n < N \end{cases} \tag{17}$$

where $\mathbf{X}_{t_0} \sim \mathcal{N}(\mathbf{0}, \mathbf{I})$; $\hat{\mathbf{X}}_{t_N}$ is approximated by $\hat{\mathbf{X}}_{t_N}^{(0)}(t_{M-1})$ at time $t_{M-1}$ using Eq. (16); for $M = 0$, we follow Shih et al. (2024a) to initialize all the samples with random noises.

**Sliding Window.** ParaSovler requires maintaining $N + 1$ parallel denoised points $\{\hat{\mathbf{X}}_{t_n}^{(k)} : n = 0, \cdots, N\}$ throughout time, which can be excessively large to feed into GPU memory when dealing with a sizable $N$. To tackle this, we use the sliding window technique from Shih et al. (2024a). This technique conducts parallel iteration merely on a subset of points $\{\hat{\mathbf{X}}_{t_n}^{(k)} : n = 0, \cdots, p - 1\}$ within a window of size $p$, where $1 \leqslant p \leqslant N$. This window size can be adjusted to adhere to the limitations of the GPU memory. The window is then promptly advanced once the states of the current timesteps within it converge.

Algorithm 1 details the complete process of the proposed ParaSolver. After preparing an array of initial guesses $\{\hat{\mathbf{X}}_{t_n}^{(0)} : n = 0, \cdots, N\}$ via random noises or a few preconditioning steps at negligible

---

**Algorithm 1:** ParaSolver: a hierarchical parallel sampling method for diffusion models

---

**Input** : Diffusion model $\boldsymbol{S}_\theta$, subinterval number $N$, preconditioning steps $M$, tolerance $\delta$, batch window size $p$, sample dimension $D$

**Output :** A sample.

1   Initialize $\{\hat{\mathbf{X}}_{t_n}^{(0)}, n = 0, \cdots, p\}$ by Eq. (17)        // Initialize with a few sampling steps

2   $n, k \leftarrow 0, 0, \; k \in [0, K],$ and $n \in [0, N-1]$

3   **while** $n < N$ **do**

4      $\Phi(t_{i+1}, t_i, \hat{\mathbf{X}}_{t_i}^{(k)}), \forall i \in \{n, \cdots, n+p-1\}$        // Solve each subproblems concurrently.

5      $\Delta_{t_i}^{(k)} \leftarrow \Phi(t_{i+1}, t_i, \hat{\mathbf{X}}_{t_i}^{(k)}) - \hat{\mathbf{X}}_{t_i}^{(k)}, \forall i \in \{n, \cdots, n+p-1\}$     // Calculate drifts.

6      $\hat{\mathbf{X}}_{t_{i+1}}^{(k+1)} \leftarrow \hat{\mathbf{X}}_{t_n}^{(k)} + \sum_{j=n}^{i} \Delta_{t_j}^{(k)}, \forall i \in \{n, \cdots, n+p-1\}$     // Update previous states via Eq. (13)

7      $s \leftarrow \underset{j}{\arg\min} \, t_j \in \left\{t_i; \hat{\mathbf{X}}_{t_i}^{(k+1)} \text{ unsatisfying Eq. (14)}, \forall i \in \{n, \cdots, n+p-1\}\right\}$   // The sliding stride

8      Obtain $\hat{\mathbf{X}}_{t_N}^{(k)}(t_{n+p-1})$ with the score from Line 4 at $t_{n+p-1}$ by Eq. (16)     // Predict clean sample.

9      $\hat{\mathbf{X}}_{t_{i+1}}^{(k+1)} \sim q(\hat{\mathbf{X}}_{t_{i+1}}^{(k+1)} | \hat{\mathbf{X}}_{t_i}^{(k+1)}, \hat{\mathbf{X}}_{t_N}^{(k)}(t_{n+p-1})), \forall i \in \{n+p, \cdots, n+p+s-1\}$     // Initialize new points.

10     $n \leftarrow n+s, \quad k \leftarrow k+1, \quad p \leftarrow \min(p, N-n)$

**Return:** $\hat{\mathbf{X}}_{t_N}^{(K)}$

---

costs (Line 1), ParaSolver initiates the parallel sampling loop at Line 2 in which a batch of samples $\{\hat{\mathbf{X}}_{t_n}^{(k)} : n = 0, \cdots, p-1\}$ within a sliding window undergo parallel denoising. Line 4 executes sampling for all the subproblems coinstantaneously in which each subproblem can be solved by existing parallel or sequential sampling methods, starting from the preceding states $\{\hat{\mathbf{X}}_{t_n}^{(k)} : n = 0, \cdots, p-1\}$. The drifts are computed in Line 5 subsequent to subproblems sampling, and Line 6 updates the previous states from the prior iteration. Line 7 checks the variation between new states and the current states and then determines the extent to which the window can slide forward. Line 9 initializes $s$ new points outside the current window according to the sliding stride.

## 5 EXPERIMENT

We evaluate ParaSolver across various high-dimensional image generation models such as latent-space diffusion model StableDiffusion-v2 (Rombach et al., 2022) and pixel-space diffusion model LSUN Church (Yu et al., 2015). The results of these experiments demonstrate that ParaSolver enhances the efficiency of sequential sampling methods by approximately $2 \sim 12$ times, all while maintaining consistent sample quality as measured by metrics like FID score or CLIP score.

### 5.1 EXPERIMENT SETTINGS

**Evaluation Metrics.** We explore five widely used metrics for our proposed PararSolver: the number of function evaluations (NFE), the iteration number (iters), wall-clock time, FID score (Heusel et al., 2017), and CLIP score (Hessel et al., 2021).

**Datasets and Models.** As per ParaDiGMS Shih et al. (2024a), we analyze our ParaSolver across latent-space and pixel-space models. For latent-space models, we leverage the StableDiffusion-v2 model model and evaluate FID and CLIP scores on a random selection of 5000 samples from the ImageNet-1k dataset . The StableDiffusion-v2 model generates images at a resolution of 768 by 768 pixels, utilizing a diffusion model that operates within a latent space dimension of 4 by 96 by 96. For pixel-space models, we employ models pretrained on the LSUN Church dataset . This LSUN Church pre-trained model operates in the pixel space with a resolution of 256 by 256. We compute the FID score on 5000 random samples of LSUN Church datasets. . We assess the performance of all methods on 8 NVIDIA RTX 3090 GPUs, each equipped with 24268 MB of memory.

**Algorithms**. We apply our approach to accelerate the performance of the state-of-the-art sequential sampling methods: DDPM (Ho et al., 2020), DDIM (Song et al., a), and DPMSolver (Lu et al., 2022). We evaluate the parallel efficiency of our ParaSolver against ParaDiGMS in accelerating the aforementioned three sequential methods. ParaDiGMS is implemented based on the the Diffusers library, and the other algorithms are all accessible in the widely-used library Diffusers (von Platen

Table 1: Quantitative comparisons of different methods on Stable Diffusion-v2 over 5000 random samples from the ImageNet-1k dataset, with classifier guidance w = 7.5. The visual comparisons are shown in Figs. 3, 4, and 2 in Section F in *Appendix*. The best results are highlighted in **bold**. "↑" (resp. "↓") means the larger (resp. smaller), the better.

| Steps | Method | Stable Diffusion-v2 | | | | | |
| | | Iters ↓ | NFE ↓ | CLIP↑ | FID↓ | Time (s)↓ | Speedup↑ |
|---|---|---|---|---|---|---|---|
| 1000 | DDPM | 1000 | 1000 | 25.6 | 55.9 | 128.0 | 1.0× |
| | DDPM + ParaDiGMS | 65 | 2024 | 25.6 | 55.6 | 32.8 | 3.9× |
| | DDPM + ParaSolver | 32 | 1065 | 25.6 | 55.3 | **15.9** | **8.1×** |
| 50 | DDIM | 50 | 50 | 25.6 | 57.2 | 6.3 | 1.0× |
| | DDIM + ParaDiGMS | 21 | 132 | 25.6 | 56.9 | 3.3 | 1.9× |
| | DDIM + ParaSolver | 13 | 83 | 25.6 | 56.9 | **1.9** | **3.3×** |
| 25 | DDIM | 25 | 25 | 25.4 | 62.9 | 3.4 | 1.0× |
| | DDIM + ParaDiGMS | 18 | 53 | 25.4 | 62.8 | 2.9 | 1.2× |
| | DDIM + ParaSolver | 11 | 49 | 25.4 | 61.9 | **1.2** | **2.8×** |
| 50 | DPMSolver | 50 | 50 | 25.6 | 57.2 | 6.3 | 1.0× |
| | DPMSolver + ParaDiGMS | 25 | 132 | 25.6 | 57.2 | 3.3 | 1.8× |
| | DPMSolver + ParaSolver | 15 | 96 | 25.6 | 57.1 | **2.2** | **3.1×** |
| 25 | DPMSolver | 25 | 25 | 25.4 | 62.3 | 3.2 | 1.0× |
| | DPMSolver + ParaDiGMS | 15 | 81 | 25.4 | 62.3 | 2.2 | 1.5× |
| | DPMSolver + ParaSolver | 11 | 58 | 25.4 | 62.1 | **1.5** | **2.1×** |

et al., 2022); hence we utilize them directly. However, we exclude the comparison with another parallel method (Tang et al., 2024) as it has not been integrated into the Diffusers library yet.

**Hyperparameter Settings.** Following the previous settings in (Shih et al., 2024a; Tang et al., 2024), we apply our ParaSolver and ParaDiGMS to DDPM with 1000 sequential sampling steps. For DDIM and DPMSolver, we consider two settings: 25 and 50 sequential sampling steps, as they are commonly utilized and capable of producing samples of similar quality to DDPM. Besides, We tune the best tolerance for ParaDiGMS via grid search on $\{0.001, 0.005, 0.01, 0.5, 0.1\}$. We find for StableDiffusion-v2, ParaDiGMS for DDPM, DDIM, and DPMSolver need to set the tolerance as 0.5, 0.01, and 0.01 to achieve the similar sample quality as the corresponding sequential method, respectively. For LSUN Church model, the tolerance of ParaDiGMS should be 0.5 and 0.001 for DDPM and DDIM. Moreover, for each setting, we sweep all the window size to find the best speedup for ParaDiGMS. For our ParaSolver on StableDiffusion-v2 model, we set the tolerance as 0.55, 0.05, and 0.05 for ParaSolver on DDPM, DDIM, and DPMSolver respectively. For LSUN Church model, the tolerances of ParaSolver are set as 0.55 and 0.005 for DDPM and DDIM, respectively.

## 5.2 EXPERIMENTAL RESULTS

**Latent-space diffusion model**. Table 1 shows the experimental results on DDPM, DDIM, DPM-Solver, and their parallel variants when combined with ParaSolver and ParaDiGMS, respectively. Initially, ParaSolver can expedite the performance of DDPM, DDIM, and DPMSolver without compromising FID and CLIP scores. The FID and CLIP scores are marginally superior to the baseline values, we think the employed tolerances might be relatively tight, thereby enhancing the overall outputs. Futhermore, ParaSolver achieves the best speedup, up to $8.1\times$ in terms of the wall-clock time. This is a new record for existing related parallel methods. Parallelization sacrifices computation for speed, resulting in a higher NFEs compared to the sequential methods. Nevertheless, when contrasted with ParaDiGMS, our ParaSolver notably reduces NFEs, even requiring mere 1065 NFEs for 1000 steps in DDPM. ParaSolver significantly diminishes the iteration count required for sequential methods by a factor ranging from 2.5 times to 31 times.

**Pixel-space diffusion model**. Table 2 shows the experimental outcomes regarding DDPM and DDIM alongside their parallel adaptations with ParaSolver and ParaDiGMS on the pretrained LSUN Church model. Similar to the latent space results, ParaSolver consistently surpasses existing techniques, achieving speedups ranging from 2.2 to 12.1 times while upholding sample quality comparable to that of the sequential methods. Producing a sample without any loss in quality within 4.1 seconds instead of 50.0 seconds can notably enhance the interactive experience for various generative scenarios like the human-in-the-loop applications (Bhattacharya et al., 2023).

Table 2: Quantitative comparisons of different methods on LSUN Church over 5000 samples. DPMSolver is not integrated with the LSUN model in the Diffusers library; thus we exclude it.

| Steps | Method | LSUN Church | | | | |
|---|---|---|---|---|---|---|
| | | Iters ↓ | NFE ↓ | FID↓ | Time (s)↓ | Speedup↑ |
| 1000 | DDPM | 1000 | 1000 | 12.7 | 50.0 | 1.0× |
| | DDPM + ParaDiGMS | 65 | 2082 | 12.7 | 10.8 | 4.6× |
| | DDPM + ParaSolver | 42 | 1079 | 12.7 | **4.1** | **12.1×** |
| 50 | DDIM | 50 | 50 | 15.5 | 1.8 | 1.0× |
| | DDIM + ParaDiGMS | 23 | 202 | 15.8 | 1.5 | 1.2× |
| | DDIM + ParaSolver | 14 | 73 | 15.7 | **0.8** | **2.2×** |
| 25 | DDIM | 25 | 25 | 15.6 | 1.1 | 1.0× |
| | DDIM + ParaDiGMS | 15 | 96 | 15.7 | 0.8 | 1.4× |
| | DDIM + ParaSolver | 10 | 44 | 15.7 | **0.5** | **2.2×** |

**The effect of $M$**. Table 3 investigates the impact of the preconditioning steps $M$. It is evident that a trade-off exists between $M$ and the acceleration achieved, with the optimal $M$ identified at $M = 10$. This is because increasing the number of preconditioning steps leads to a more accurate initialization but also requires additional time. As we implement the preconditioning steps through a sequential sampling approach, the efficiency of parallel processes is affected by a large $M$. We believe that by executing the preconditioning steps in parallel, we can enhance the precision of initialization, facilitating a faster acceleration even with a higher $M$.

Table 3: The effect of preconditioning steps $M$ using LSUN Church over 5000 random samples. Sect. H in *Appendix* shows the visual comparisons.

| Steps | Method | $M$ | LSUN Church | | | | |
|---|---|---|---|---|---|---|---|
| | | | Iters↓ | NFE↓ | FID↓ | Time (s)↓ | Speedup↑ |
| 1000 | DDPM | - | 1000 | 1000 | 12.7 | 50.0 | 1.0× |
| | DDPM + ParaSolver | 0 | 54 | 1036 | 12.8 | 4.4 | 11.3× |
| | DDPM + ParaSolver | 1 | 54 | 1017 | 12.8 | 4.3 | 11.6× |
| | DDPM + ParaSolver | 10 | 42 | 1066 | 12.7 | **4.1** | **12.1×** |
| | DDPM + ParaSolver | 15 | 34 | 1078 | 12.7 | 4.2 | 11.9× |

**The effect of $N$**. Table 4 examines the influence of $N$. The most favorable wall-clock time is observed at $N = 15$. At $N = 1$, ParaSolver reverts to a sequential sampling approach. However, for $N = 5$ and $N = 10$, the wall-clock time exceeds the sequential sampling because the current implementation for solving the subproblems is through the sequential sampling. With each parallel iteration involving the sequential sampling, the parallel efficiency is significantly compromised. Substituting the sequential sampling with parallel techniques could substantially enhance the efficiency. We believe that by leveraging optimized implementations, the wall-clock times in Table 4 can be further refined. For $N = 30$, ParaSolver aligns with existing parallel methods with our initialization method, necessitating the computation of the entire triangular part as specified in Eq. (5). This demanding computational task results in extended wall-clock times. We note that to tune the window size, the wall-clock time in the Table 4 can be further improved. Moreover, the visual comparisons presented in Section G in the *Appendix* indicate that employing a larger value of $N$ may result in increased errors on the refining samples during parallel iteration. This occurs because the ultimate clean samples are connected to all samples from earlier time steps, causing their errors dur-

Table 4: The effect of the number of subintervals $N$ using LSUN Church over 5000 random samples. Our ParaSolver is employed to expedite DDIM with 30 sequential steps. We set $M = 10$ and fix the window size $p = 30$. The visual comparisons are shown in Section G in *Appendix*.

| Method | $N = 1$ | | $N = 5$ | | $N = 10$ | | $N = 15$ | | $N = 30$ | |
|--------|------|------|------|------|------|------|------|------|------|------|
| | FID↓ | time (s)↓ | FID↓ | time (s)↓ | FID↓ | time (s)↓ | FID↓ | time (s)↓ | FID↓ | time (s)↓ |
| Ours | 15.4 | 1.2 | 15.3 | 1.8 | 15.3 | 1.4 | 15.2 | **0.9** | 15.2 | 1.5 |

Table 5: The effect of the number of GPUs using LSUN Church. Our ParaSolver is employed to expedite DDPM with 1000 sequential steps.

| Method | 1 GPU | | 2 GPUs | | 4 GPUs | | 8 GPUs | |
|--------|---------|----------|---------|----------|---------|----------|---------|----------|
| | Time (s)↓ | Speedup↑ | Time (s)↓ | Speedup↑ | Time (s)↓ | Speedup↑ | Time (s)↓ | Speedup↑ |
| DDPM | 50.0 | 1.0× | 50.0 | 1.0× | 50.0 | 1.0× | 50.0 | 1.0× |
| ParaDiGMS | 37.0 | 1.4× | 23.3 | 2.1× | 15.2 | 3.3× | 10.8 | 4.6× |
| Ours | **19.7** | **2.5×** | **13.0** | **3.8×** | **9.9** | **5.1×** | **4.1** | **12.1×** |

ing updation to all accumulate in the final clean samples. Refining these errors requires additional parallel iterations, which can reduce the parallel efficiency.

**The effect of the number of GPUs**. In Table 5, the analysis focuses on how the number of GPUs affects performance. When ample computing resources are available, the parallel efficiency of our method can be greatly released than ParaDiGMS, resulting in a speedup of 12.1 times. Additionally, even in scenarios with limited computing power, such as utilizing only a single RTX 3090 GPU, the speedup can reach 2.5 times, whereas ParaDiGMS shows only modest improvement. This is attributed to our sampling process, which only needs to handle a sparse banded NEs system, in contrast to ParaDiGMS, which requires full computation over the dense triangular NEs system.

Table 6: The effect of the tolerance $\delta$ on FID using LSUN Church over 5000 random samples. Our ParaSolver is employed to DDIM with 30 sequential steps with FID 15.5.

| $M$ | $\delta$ | 0.001 | 0.005 | 0.01 | 0.05 | 0.1 |
|-----|-----|-------|-------|------|------|-----|
| 1 | FID | 15.3 | 15.6 | 21.4 | 40.6 | 104.6 |
| 5 | FID | 14.8 | 15.3 | 20.7 | 36.4 | 50.3 |
| 10 | FID | 14.3 | 15.2 | 16.6 | 18.2 | 23.2 |

**The effect of the tolerance** $\delta$. In Table 6, we demonstrate how tolerance impacts FID under different preconditioning steps $M$. It is evident that ParaSolver can uphold a similar or better FID to the baseline when $\delta = 0.005$ and $\delta = 0.001$, respectively. Furthermore, a higher $M$ leads to a better FID even under a looser tolerance.

**The effect of the window size** $p$. In Figure 1, we examine how the window size $p$ affects the speedup of 1000 DDPM over the LSUN Church Model. The peak speedup is observed at $p = 36$ rather than the maximal $p$. This is because the parallel efficiency is inundated by the significant computational burden associated with larger window sizes. In situations with restricted computational resources, it is crucial to select an appropriate window size for optimal performance.

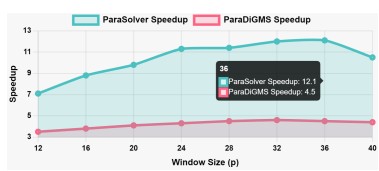

Figure 1: The effect of $p$

## 6 CONCLUSION

In this paper, we conceptualized the process of sequential sampling in DPMs as solving a system of banded NEs leading to a general parallel sampling approach . Leveraging the semi-linear structure of the Jacobian matrix within this NEs system, we derived a more precise parallel sampling iteration. Building upon this iteration, we introduced a novel hierarchical parallel sampling technique called ParaSolver. ParaSolver is computationally efficient and seamlessly compatible with existing parallel and sequential methodologies. Extensive experiments demonstrate that our ParaSolver can achieve a speedup of up to **12×** in terms of wall-clock time without degrading the sample quality.

## ETHICS & REPRODUCIBILITY STATEMENTS

The proposed sampling algorithm is designed to accelerate the pre-trained models. We have read and adhere to the Code of Ethics of ICLR 2025. Thus, no further additional information on human subjects and potentially harmful insights is involved. Moreover, we set a random seed of **1** during the experiment, which ensures reproducibility. We will make the source code publicly available on Github.

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

C­ONTENTS

## A  P­ROOF OF P­ROPOSITION 1

Firstly, it is straightforward to confirm that the analytical expression $\Phi(t_{n+1}, t_n, \boldsymbol{Y}_n)$ possesses a unique solution. Now, based on the principle of proof by contradiction, suppose that there exist two distinct solutions $\mathbf{A}_{t_0}, \mathbf{A}_{t_1}, \cdots, \mathbf{A}_{t_N}$ and $\mathbf{B}_{t_0}, \mathbf{B}_{t_1}, \cdots, \mathbf{B}_{t_N}$, so they satisfy:

$$\begin{cases} \mathbf{A}_{t_{n+1}} = \Phi(t_{n+1}, t_n, \mathbf{A}_{t_n}) \\ \mathbf{B}_{t_{n+1}} = \Phi(t_{n+1}, t_n, \mathbf{B}_{t_n}) \end{cases} \tag{18}$$

Assume by induction that for $0 \leqslant n \leqslant N$, $\mathbf{A}_{t_n} = \mathbf{B}_{t_n}$, so this yields

$$\mathbf{A}_{t_{n+1}} = \Phi(t_{n+1}, t_n, \mathbf{A}_{t_n}) = \Phi(t_{n+1}, t_n, \mathbf{B}_{t_n}) \tag{19}$$

Thus, the two sets of solutions $\mathbf{A}_{t_0}, \mathbf{A}_{t_1}, \cdots, \mathbf{A}_{t_N}$ and $\mathbf{B}_{t_0}, \mathbf{B}_{t_1}, \cdots, \mathbf{B}_{t_N}$ are the identical.

Secondly, it is easy to confirm that the solutions at re unbiased estimation of the sequential sampling results. In particular, according to Eq. (4) and Eq. (3), the the sequential sampling results can be expressed as

$$\Phi(t_{n+1}, 0, \mathbf{X}_0) = \mathbf{X}_0 + \int_0^{t_n} \varphi(\mathbf{X}_\tau, \tau) d\tau + \int_0^{t_n} \sigma_\tau d\mathbf{W}$$

$$= \mathbf{X}_0 + \sum_{i=0}^n \int_{t_i}^{t_{i+1}} \varphi(\mathbf{X}_\tau, \tau) d\tau + \sum_{i=0}^n \int_{t_i}^{t_{i+1}} \sigma_\tau d\mathbf{W}$$

$$= \sum_{i=0}^n \mathbf{X}_{t_i} + \sum_{i=0}^n \int_{t_i}^{t_{i+1}} \varphi(\mathbf{X}_\tau, \tau) d\tau + \sum_{i=0}^n \int_{t_i}^{t_{i+1}} \sigma_\tau d\mathbf{W} \qquad (20)$$

$$= \sum_{i=0}^n \Phi(t_{i+1}, t_i, \mathbf{X}_{t_i})$$

Therefore, we have

$$\mathrm{E}[\mathbf{X}_{t_{n+1}}] = \mathrm{E}[\Phi(t_{n+1}, 0, \mathbf{X}_0)] = \sum_{i=0}^n \mathrm{E}[\mathbf{X}_{t_i}] + \sum_{i=0}^n \int_{t_i}^{t_{i+1}} \mathrm{E}[\varphi(\mathbf{X}_\tau, \tau)] d\tau + \sum_{i=0}^n \int_{t_i}^{t_{i+1}} \sigma_\tau d\mathbf{W} \qquad (21)$$

For our method, we have

$$\mathrm{E}[\mathbf{A}_{t_{n+1}}] = \mathrm{E}[\Phi(t_{n+1}, t_n, \mathbf{A}_{t_n})]$$

$$= \mathrm{E}[\mathbf{A}_{t_n}] + \int_{t_n}^{t_{n+1}} \mathrm{E}[\varphi(\mathbf{A}_\tau, \tau)] d\tau + \int_{t_n}^{t_{n+1}} \sigma_\tau d\mathbf{W} \qquad (22)$$

$$= \sum_{i=0}^n \mathrm{E}[\mathbf{A}_{t_i}] + \sum_{i=0}^n \int_{t_i}^{t_{i+1}} \mathrm{E}[\varphi(\mathbf{A}_\tau, \tau)] d\tau + \sum_{i=0}^n \int_{t_i}^{t_{i+1}} \sigma_\tau d\mathbf{W}$$

Recall that $\mathbf{A}_{t_0}$ and $\mathbf{X}_0$ follow the same distribution. Therefore, according to Eq. (21) and Eq. (22), it is easy to see that $\mathrm{E}[\mathbf{A}_{t_n}] = \mathrm{E}[\mathbf{X}_{t_n}]$ because of the linear nature of expectation.

## B    PROOF OF PROPOSITION 2

Before proceeding with the proof, we need to state several assumptions and definitions. Following the Banach fixed-point theorem Banach (1922), we first make the following assumptions.

**Assumption 1.** *The denoising network* $\mathbf{S}_\theta(\mathbf{X}_t, t)$ *is continuous and differentiable. Holding t fixed and letting θ the optimal model θ\*, the mapping operator* $\mathbf{S}_{\theta*}(\cdot, t)$ *is a contraction, i.e., there exists a constant* $0 \leqslant \gamma < 1$, *such that:*

$$\|\mathbf{S}_{\theta*}(\mathbf{U}, t) - \mathbf{S}_{\theta*}(\mathbf{V}, t)\| \leqslant \gamma \|\mathbf{U} - \mathbf{V}\|, \forall \mathbf{U}, \mathbf{V} \in \mathbb{R}^{HWC} \qquad (23)$$

***Remark:*** It is sensible to consider the optimal denoising network $\mathbf{S}_{\theta*}(\mathbf{X}_t, t)$ as a contraction mapping, as it can convert data points from a large space into a single dense data space that adheres to a standard Gaussian distribution, thereby effectively compressing the data points in the larger space.

For simplicity of expression, we denote $\mathcal{G}_n = \frac{\partial \Phi(t_{n+1}, t_n, \hat{\mathbf{X}}_{t_n})}{\partial \hat{\mathbf{X}}_{t_n}}$ and $\lambda_{t_n} = log \frac{\alpha_{t_n}}{\sigma_{t_n}}$ (one half of the log-SNR). Recall that we define $\mathbf{X}_{t_0}$ and $\mathbf{X}_{t_N}$ as the random noise and clean image respectively. Besides the forward process $p(\mathbf{X}_{t_n}|\mathbf{X}_{t_N}) = \mathcal{N}(\alpha_{t_n} \mathbf{X}_{t_N}, \sigma_{t_n} \mathbf{I})$ where $\alpha$ and $\sigma$ are the noise schedule. Then it is easy to know that $\lambda_{t_n}$ is a strictly decreasing function of $t$.

Following Proposition 3.1 in Lu et al. (2022), we can obtain the exact solution of diffusion SDEs, i.e.,

$$\Phi(t, s, \mathbf{X}_s) = \frac{\alpha_t}{\alpha_s} \mathbf{X}_s - \alpha_t \int_{\lambda_s}^{\lambda_t} e^{-\lambda} \mathbf{S}_{\theta*}(\mathbf{X}_\lambda, \lambda) d\lambda + \int_{\lambda_s}^{\lambda_t} g(\lambda) d\mathbf{W}. \qquad (24)$$

Now we conduct the proof. Recall that we have the residual function:

$$\mathcal{R}_{t_n} = \hat{\mathbf{X}}_{t_{n+1}} - \Phi(t_{n+1}, t_n, \hat{\mathbf{X}}_{t_n}). \tag{25}$$

Since $\hat{\mathbf{X}}_{t_{n+1}}$ is linear for $\mathcal{R}_{t_n}$, it is sufficiently smooth. We now need to affirm whether the nonlinear part $\Phi(t_{n+1}, t_n, \hat{\mathbf{X}}_{t_n})$ is sufficiently smooth. It is well known that if the normal of Jacobian matrix of $\Phi(t_{n+1}, t_n, \hat{\mathbf{X}}_{t_n})$ is bounded by a small number, then the function is smooth enough. In light of this, we seek to find the upper bound of the norm of the Jacobian matrix $\mathcal{G}_n = \frac{\partial \Phi(t_1, t_0, \hat{\mathbf{X}}_{t_n})}{\partial \hat{\mathbf{X}}_{t_n}}$.

According to the definition of gradient, we get:

$$\|\mathcal{G}_n\| \geqslant \lim_{\mathbf{h} \to 0} \frac{\|\Phi(t_{n+1}, t_n, \hat{\mathbf{X}}_{t_n} + \mathbf{h}) - \Phi(t_{n+1}, t_n, \hat{\mathbf{X}}_{t_n})\|}{\|\mathbf{h}\|} \tag{26}$$

The equation holds with $\mathbf{h}$ being on the principle direction of singular value decomposition of $\mathcal{G}$. Here we donote such subset of $\mathbf{h}$ as $\mathbf{h}'$. Then, we have

$$\|\mathcal{G}_n\| = \lim_{\mathbf{h}' \to 0} \frac{\|\Phi(t_{n+1}, t_n, \hat{\mathbf{X}}_{t_n} + \mathbf{h}') - \Phi(t_{n+1}, t_n, \hat{\mathbf{X}}_{t_n})\|}{\|\mathbf{h}'\|} \tag{27}$$

Utilizing Eq. (24) yields:

$$
\begin{aligned}
&\|\Phi(t_{n+1}, t_n, \hat{\mathbf{X}}_{t_n} + \mathbf{h}') - \Phi(t_{n+1}, t_n, \hat{\mathbf{X}}_{t_n})\| \\
&= \left\| \frac{\alpha_{t_{n+1}}}{\alpha_{t_n}} \mathbf{h}' - \alpha_{t_{n+1}} \int_{\lambda_{t_n}}^{\lambda_{t_{n+1}}} e^{-\lambda} [\mathbf{S}_{\theta*}(\mathbf{X}_\lambda + \mathbf{h}', \lambda) - \mathbf{S}_{\theta*}(\mathbf{X}_\lambda, \lambda)] d\lambda \right\| \\
&\leqslant \frac{\alpha_{t_{n+1}}}{\alpha_{t_n}} \|\mathbf{h}'\| + \alpha_{t_{n+1}} \int_{\lambda_{t_n}}^{\lambda_{t_{n+1}}} e^{-\lambda} \left\| \mathbf{S}_{\theta*}(\mathbf{X}_\lambda + \mathbf{h}', \lambda) - \mathbf{S}_{\theta*}(\mathbf{X}_\lambda, \lambda) \right\| d\lambda \\
&\leqslant \frac{\alpha_{t_{n+1}}}{\alpha_{t_n}} \|\mathbf{h}'\| + \alpha_{t_{n+1}} \gamma \|\mathbf{h}'\| \int_{\lambda_{t_n}}^{\lambda_{t_{n+1}}} e^{-\lambda} d\lambda \\
&= \frac{\alpha_{t_{n+1}}}{\alpha_{t_n}} \|\mathbf{h}'\| + \sigma_{t_{n+1}} \gamma \|\mathbf{h}'\| (e^{\lambda_{t_{n+1}} - \lambda_{t_n}} - 1) \\
&= \frac{\alpha_{t_{n+1}}}{\alpha_{t_n}} \|\mathbf{h}'\| + \sigma_{t_{n+1}} \gamma \|\mathbf{h}'\| (e^{log \frac{\alpha_{t_{n+1}} \sigma_{t_n}}{\alpha_{t_n} \sigma_{t_{n+1}}}} - 1).
\end{aligned} \tag{28}
$$

Taking Eq. (28) into Eq. (27), we obtain the upper bound of the gradient as follows:

$$\|\mathcal{G}_n\| \leqslant \frac{\alpha_{t_{n+1}}}{\alpha_{t_n}} + \sigma_{t_{n+1}} \gamma (e^{log \frac{\alpha_{t_{n+1}} \sigma_{t_n}}{\alpha_{t_n} \sigma_{t_{n+1}}}} - 1) \tag{29}$$

$$\leqslant \frac{\alpha_{t_{n+1}}}{\alpha_{t_n}} + \sigma_{t_{n+1}} \gamma (\frac{\alpha_{t_{n+1}} \sigma_{t_n}}{\alpha_{t_n} \sigma_{t_{n+1}}} - 1). \tag{30}$$

The relationship between the Jacobian norm and the noise schedule parameters $\alpha$ and $\sigma$ is clearly illustrated in Eq. (29). Given that $\alpha$ and $\sigma$ in DPMs are strictly monotonically decreasing and increasing with respect to $t$, respectively, we can conclude that the ratios $\frac{\alpha_{t_{n+1}}}{\alpha_{t_n}}$ and $\frac{\sigma_{t_n}}{\sigma_{t_{n+1}}}$ are nearly equal to 1, as their rates of change with respect to the time step are minimal. As a result, $\|\mathcal{G}_n\|$ is constrained to a small value close to 1, indicating a highly smooth function $\Phi(t_{n+1}, t_n, \hat{\mathbf{X}}_{t_n})$, which implies that the residual $\mathcal{R}_{t_n}$ is adequately smooth.

## C  PROOF OF PROPOSITION 3

According to Definition 1, the specific form of the residual term for our banded NEs system is as follows:

$$\mathcal{R}_{t_0:t_N}^{(k)} = \begin{cases} \hat{\mathbf{X}}_{t_0}^{(0)} - \mathbf{X}_{t_0}, \\ \hat{\mathbf{X}}_{t_{n+1}}^{(k+1)} - \Phi(t_{n+1}, t_n, \hat{\mathbf{X}}_{t_n}^{(k)}), & \text{if } 0 \leqslant n \leqslant N - 1 \end{cases} \tag{31}$$

Using this specific form, we can express the Newton update $(J_{t_0:t_N}^{(k)})^{-1}\mathcal{R}_{t_0:t_N}^{(k)}$ in a particular manner

$$
\begin{bmatrix}
\boldsymbol{I} & & & & \\
-\dfrac{\partial \Phi(t_1,t_0,\hat{\mathbf{X}}_{t_0}^{(k)})}{\partial \hat{\mathbf{X}}_{t_0}^{(k)}} & \boldsymbol{I} & & & \\
& -\dfrac{\partial \Phi(t_2,t_1,\hat{\mathbf{X}}_{t_1}^{(k)})}{\partial \hat{\mathbf{X}}_{t_1}^{(k)}} & \ddots & & \\
& & \ddots & & \\
& & & -\dfrac{\partial \Phi(t_N,t_{N-1},\hat{\mathbf{X}}_{t_{N-1}}^{(k)})}{\partial \hat{\mathbf{X}}_{t_{N-1}}^{(k)}} & \boldsymbol{I}
\end{bmatrix}^{-1}
\left(
\begin{array}{c}
\hat{\mathbf{X}}_{t_0}^{(k)} - \mathbf{X}_{t_0} \\
\hat{\mathbf{X}}_{t_1}^{(k)} - \Phi(t_1,t_0,\hat{\mathbf{X}}_{t_0}^{(k)}) \\
\hat{\mathbf{X}}_{t_2}^{(k)} - \Phi(t_2,t_1,\hat{\mathbf{X}}_{t_1}^{(k)}) \\
\vdots \\
\hat{\mathbf{X}}_{t_N}^{(k)} - \Phi(t_N,t_{N-1},\hat{\mathbf{X}}_{t_{N-1}}^{(k)})
\end{array}
\right).
\tag{32}
$$

Rearranging Eq. (32) by multiplying through by the Jacobian and taking into the Eq. (9), we derive the fundamental recurrence used for our parallel iteration at iteration $k$.

$$
\begin{cases}
\hat{\mathbf{X}}_{t_0}^{(k+1)} = \hat{\mathbf{X}}_{t_0}^{(0)} = \mathbf{X}_{t_0} \sim \mathcal{N}(\mathbf{0},\boldsymbol{I}), \\
\hat{\mathbf{X}}_{t_{n+1}}^{(k+1)} = \Phi(t_{n+1},t_n,\hat{\mathbf{X}}_{t_n}^{(k)}) + \dfrac{\partial \Phi(t_{n+1},t_n,\hat{\mathbf{X}}_{t_n}^{(k)})}{\partial \hat{\mathbf{X}}_{t_n}^{(k)}}\left(\hat{\mathbf{X}}_{t_n}^{(k+1)} - \hat{\mathbf{X}}_{t_n}^{(k)}\right).
\end{cases}
\tag{33}
$$

# D    PROOF OF PROPOSITION 4

**Assumption 2.** *The residual $\mathcal{R}_{t_0:t_N}$ is $L_1$-Lipschitz continuous with respect to input $\hat{\mathbf{X}} := (\hat{\mathbf{X}}_{t_0},\cdots,\hat{\mathbf{X}}_{t_{N-1}})$, satisfying:*

$$
\|\mathcal{R}_{t_0:t_N}(\mathbf{U}) - \mathcal{R}_{t_0:t_N}(\mathbf{V})\| \leq L_1\|\mathbf{U}-\mathbf{V}\|, \forall \mathbf{U},\mathbf{V} \in \mathbb{R}^{(N+1)\times HWC}
\tag{34}
$$

**Assumption 3.** *Suppose that the Hessian matrix $H(\hat{\mathbf{X}})$ of the residual $\mathcal{R}_{t_0:t_N}$ with respect to input $\hat{\mathbf{X}}$ is bounded, that is, there is a constant $L_2$ such that*

$$
\|H(\hat{\mathbf{X}})\| \leq L_2, \forall \hat{\mathbf{X}} \in \mathbb{R}^{(N+1)\times HWC}
\tag{35}
$$

*Remark:* Proposition 2 demonstrates that the residual $\mathcal{R}_{t_0:t_N}$ is sufficiently and possesses a very small upper bound on the Jacobian of the nonlinear part $\Phi(t_n,t_{n-1},\hat{\mathbf{X}}_{t_{n-1}})$, suggesting that $L_1$ and $L_2$ are small real numbers. This insight will aid in analyzing the convergence behavior of the proposed update rule in Eq. (11).

For the sake of simplicity, we denote $\mathcal{G}_n = \dfrac{\partial \Phi(t_{n+1},t_n,\hat{\mathbf{X}}_{t_n})}{\partial \hat{\mathbf{X}}_{t_n}}$; $\mathcal{R}(\hat{\mathbf{X}}) = \mathcal{R}_{t_0:t_N}(\hat{\mathbf{X}})$. Therefore, according to the Jacobian in Eq. (32), the Jacobian matrix for $\mathcal{R}(\hat{\mathbf{X}})$ has the following convenient form:

$$
\mathcal{J} = \begin{bmatrix}
\boldsymbol{I} & & & \\
-\mathcal{G}_0 & \boldsymbol{I} & & \\
& \ddots & \ddots & \\
& & -\mathcal{G}_{N-1} & \boldsymbol{I}
\end{bmatrix}.
\tag{36}
$$

Since we use identity matrix $I$ to substitute $\mathcal{G}_n$, we thus have an approximated matrix for $\mathcal{J}$, denoted as,

$$
\tilde{\mathcal{J}} = \begin{bmatrix}
\mathbf{I} & & & \\
-\mathbf{I} & \mathbf{I} & & \\
& \ddots & \ddots & \\
& & -\mathbf{I} & \mathbf{I}
\end{bmatrix}.
\tag{37}
$$

**Proof of a descent direction.** Now, we first prove that the approximated Jacobian matrix $\tilde{\mathcal{J}}$ is a descent direction for $\mathcal{R}(\hat{\mathbf{X}})$, which is not contradictory to the descent direction of the true Jacobian matrix $\mathcal{J}$. Formally, to complete the proof, it suffices to show their inner product satisfies that:

$$\langle \mathcal{J}, \tilde{\mathcal{J}} \rangle > 0, \forall \hat{\mathbf{X}} \in \mathbb{R}^{(N+1) \times HWC}. \tag{38}$$

According to the relation between the trace and Frobenius inner product of the matrix, we have

$$
\begin{aligned}
\langle \mathcal{J}, \tilde{\mathcal{J}} \rangle &= tr(\mathcal{J}^\top \tilde{\mathcal{J}}) \\
&= tr\left(
\begin{bmatrix}
\mathbf{I} + \mathcal{G}_0^\top & -\mathcal{G}_0^\top & & & \\
-\mathbf{I} & \mathbf{I} + \mathcal{G}_1^\top & -\mathcal{G}_1^\top & & \\
& \ddots & \ddots & \ddots & \\
& & -\mathbf{I} & \mathbf{I} + \mathcal{G}_{N-1}^\top & -\mathcal{G}_{N-1}^\top \\
& & & -\mathbf{I} & \mathbf{I}
\end{bmatrix}
\right) \\
&= \sum_{n=0}^{N} tr(\mathbf{I}) + \sum_{n=0}^{N-1} tr(\mathcal{G}_n) \\
&\overset{(a)}{\geqslant} (N+1)HWC - \sqrt{HWC} \sum_{n=0}^{N-1} \left\| \mathcal{G}_n \right\|
\end{aligned}
\tag{39}
$$

Here, the inequation (a) uses the Lemma 1. According to Proposition 2, we know $\left\| \mathcal{G}_n \right\|$ is upper bounded by a number close to 1. Therefore, Eq. (40) becomes

$$\langle \mathcal{J}, \tilde{\mathcal{J}} \rangle \geqslant \quad (N+1)HWC - N\sqrt{HWC} > 0 \tag{40}$$

**Proof of convergence speed.** In this part, we will prove that the convergence speed of the update rule in Eq. (13) falls between linear and quadratic convergence. According to Taylor's theorem, any function that has a continuous second derivative can be represented by an expansion about a point that is close to a root of this function. In view of this, we suppose a root of the residual function $\mathcal{R}(\hat{\mathbf{X}})$ is $\mathbf{A}$. Then the expansion of $\mathcal{R}(\mathbf{A})$ on point $\hat{\mathbf{X}}^{(k)}$ at parallel iteration $k$ is:

$$\mathcal{R}(\mathbf{A}) = \mathcal{R}(\hat{\mathbf{X}}^{(k)}) + \mathcal{J}^\top (\mathbf{A} - \hat{\mathbf{X}}^{(k)}) + \underbrace{\frac{1}{2}(\mathbf{A} - \hat{\mathbf{X}}^{(k)})^\top \mathbf{H}(\xi)(\mathbf{A} - \hat{\mathbf{X}}^{(k)})}_{\mathbf{R}}, \tag{41}$$

where the Lagrange form of the Taylor series expansion remainder is denoted as $\mathbf{R}$ in which $\xi$ is in between $\hat{\mathbf{X}}^{(k)}$ and $\mathbf{A}$. And $\mathbf{H}(\cdot)$ is the Hessian matrix with respect to $\hat{\mathbf{X}}^{(k)}$. Since $\mathbf{A}$ is the root, Eq. (41) becomes:

$$\mathbf{0} = \mathcal{R}(\hat{\mathbf{X}}^{(k)}) + \mathcal{J}^\top (\mathbf{A} - \hat{\mathbf{X}}^{(k)}) + \mathbf{R}. \tag{42}$$

Multiply both sides by the inverse of $\mathcal{J}^\top$ and rearranging gives:

$$[\mathcal{J}^\top]^{-1} \mathcal{R}(\hat{\mathbf{X}}^{(k)}) + \mathbf{A} - \hat{\mathbf{X}}^{(k)} = -[\mathcal{J}^\top]^{-1} \mathbf{R}. \tag{43}$$

Remembering that the update rule in Eq. (12) uses $\tilde{\mathcal{J}}$ to approximate $\mathcal{J}$, yielding:

$$
\begin{aligned}
\hat{\mathbf{X}}^{(k+1)} &= \hat{\mathbf{X}}^{(k)} - \tilde{\mathcal{J}}^{-1} \mathcal{R}(\hat{\mathbf{X}}^{(k)}) \\
&= \hat{\mathbf{X}}^{(k)} - [\mathcal{J}^\top]^{-1} \mathcal{R}(\hat{\mathbf{X}}^{(k)}) + \left\{ [\mathcal{J}^\top]^{-1} - \tilde{\mathcal{J}}^{-1} \right\} \mathcal{R}(\hat{\mathbf{X}}^{(k)}).
\end{aligned}
\tag{44}
$$

Combining Eq. (44) with Eq. (43), we find:

$$\mathbf{A} - \hat{\mathbf{X}}^{(k+1)} + \left\{ [\mathcal{J}^\top]^{-1} - \tilde{\mathcal{J}}^{-1} \right\} \mathcal{R}(\hat{\mathbf{X}}^{(k)}) = -[\mathcal{J}^\top]^{-1} \mathbf{R}. \tag{45}$$

Denote by $\varepsilon^{(k)} = \mathbf{A} - \hat{\mathbf{X}}^{(k)}$, Eq. (45) becomes:

$$\varepsilon^{(k+1)} = -\left\{[\mathcal{J}^\top]^{-1} - \tilde{\mathcal{J}}^{-1}\right\}\mathcal{R}(\hat{\mathbf{X}}^{(k)}) - \frac{1}{2}[\mathcal{J}^\top]^{-1}[\varepsilon^{(k)}]^\top\mathbf{H}(\xi)\varepsilon^{(k)}. \tag{46}$$

Taking the norm of both sides gives:

$$
\begin{aligned}
\left\|\varepsilon^{(k+1)}\right\| &= \left\| -\left\{[\mathcal{J}^\top]^{-1} - \tilde{\mathcal{J}}^{-1}\right\}\mathcal{R}(\hat{\mathbf{X}}^{(k)}) - \frac{1}{2}[\mathcal{J}^\top]^{-1}[\varepsilon^{(k)}]^\top\mathbf{H}(\xi)\varepsilon^{(k)}\right\| \\
&\leqslant \left\|[\mathcal{J}^\top]^{-1} - \tilde{\mathcal{J}}^{-1}\right\|\left\|\mathcal{R}(\hat{\mathbf{X}}^{(k)})\right\| + \frac{1}{2}\left\|[\mathcal{J}^\top]^{-1}[\varepsilon^{(k)}]^\top\mathbf{H}(\xi)\varepsilon^{(k)}\right\| \\
&\stackrel{(b)}{=} \left\|[\mathcal{J}^\top]^{-1} - \tilde{\mathcal{J}}^{-1}\right\|\left\|\mathcal{R}(\mathbf{A}) - \mathcal{R}(\hat{\mathbf{X}}^{(k)})\right\| + \frac{1}{2}\left\|[\mathcal{J}^\top]^{-1}[\varepsilon^{(k)}]^\top\mathbf{H}(\xi)\varepsilon^{(k)}\right\| \\
&\stackrel{(c)}{\leqslant} L_1\left\|[\mathcal{J}^\top]^{-1} - \tilde{\mathcal{J}}^{-1}\right\|\left\|\varepsilon^{(k)}\right\| + \frac{1}{2}\left\|[\mathcal{J}^\top]^{-1}[\varepsilon^{(k)}]^\top\mathbf{H}(\xi)\varepsilon^{(k)}\right\| \\
&\stackrel{(d)}{\leqslant} \underbrace{L_1\left\|[\mathcal{J}^\top]^{-1} - \tilde{\mathcal{J}}^{-1}\right\|\left\|\varepsilon^{(k)}\right\|}_{M_1} + \underbrace{\frac{L_2}{2}\left\|\mathcal{J}^{-1}\right\|\left\|\varepsilon^{(k)}\right\|^2}_{M_2}
\end{aligned}
\tag{47}
$$

here $(b)$ holds since $\mathbf{A}$ is the root of $\mathcal{R}(\hat{\mathbf{X}}^{(k)})$; $(c)$ relies on Assumption 2; $(d)$ is derived from the properties of matrix transpose and norm, as well as Assumption 3.

Since the identity matrix approximation provides a descent direction and the residual function is smooth enough, it indicates that convergence is possible, implying that $M_1$ and $M_2$ are relatively small. Therefore, the above Eq. (47) describes the rate at which the sequence $\hat{\mathbf{X}}^{(k)}$ ($\varepsilon^{(k)} = \mathbf{A} - \hat{\mathbf{X}}^{(k)}$) converges to the root $\mathbf{A}$. Since the inequality contains both linear and quadratic terms, we can conclude that the convergence rate of the sequence $\hat{\mathbf{X}}^{(k)}$ is at least linear but influenced by the quadratic term. This means that the convergence rate of the sequence $\hat{\mathbf{X}}^{(k)}$ is faster than linear but slower than quadratic. The exact order depends on the specific values of $M_1$ and $M_2$:

- If $M_1$ is small and $M_2$ is non-zero, then the quadratic term will dominate, and the sequence will converge at a quadratic rate.
- If $M_1$ is non-zero and $M_2$ is small or zero, then the linear term will dominate, and the sequence will converge at a linear rate.

Since this study mainly concentrates on parallel sampling for DPMs, offering specific values for $M$ and detailed convergence speeds falls outside our current scope. We provide this initial insight for a general understanding and will explore the specific relationship between $M_1$ and $M_2$ in future work.

# E  PROOF OF LEMMA 1

**Lemma 1.** *For any matrix $\mathbf{A} \in \mathbb{R}^n$, the relationship between the trace $tr(\mathbf{A})$ and the Frobenius norm $\|\mathbf{A}\|_F$ satisfies the following inequation:*

$$|tr(\mathbf{A})| \leqslant \|\mathbf{A}\|_F \cdot \sqrt{n}$$

**Proof:** To show the relationship $|tr(\mathbf{A})| \leqslant \|\mathbf{A}\|_F \cdot \sqrt{n}$, we can start with the definitions of both concepts.

1. Trace of a matrix $\mathbf{A}$:

$$tr(\mathbf{A}) = \sum_{i=1}^{n} a_{ii}$$

where $a_{ii}$ are the diagonal elements of the matrix $\mathbf{A}$.

2. Frobenius Norm of a matrix $\mathbf{A}$:

$$\|\mathbf{A}\|_F = \sqrt{\sum_{i=1}^{n} \sum_{j=1}^{n} |a_{ij}|^2}$$

To show the relationship $|\text{tr}(\mathbf{A})| \leqslant \|\mathbf{A}\|_F \cdot \sqrt{n}$, we can use the Cauchy-Schwarz inequality.

Using the Cauchy-Schwarz inequality in the context of sums, we have:

$$\left( \sum_{i=1}^{n} |a_{ii}| \right)^2 \leqslant \left( \sum_{i=1}^{n} 1^2 \right) \left( \sum_{i=1}^{n} |a_{ii}|^2 \right)$$

**Step 2: Simplify the Terms.** The first term simplifies to $n$:

$$\sum_{i=1}^{n} 1^2 = n$$

Thus, we can rewrite the inequality as:

$$\left( \sum_{i=1}^{n} |a_{ii}| \right)^2 \leqslant n \sum_{i=1}^{n} |a_{ii}|^2$$

The Frobenius norm can be expressed as:

$$\|\mathbf{A}\|_F^2 = \sum_{i=1}^{n} \sum_{j=1}^{n} |a_{ij}|^2$$

It includes the sum of the squares of all elements in the matrix $\mathbf{A}$, hence:

$$\|\mathbf{A}\|_F^2 \geqslant \sum_{i=1}^{n} |a_{ii}|^2$$

Combining these results, we have:

$$\left( \sum_{i=1}^{n} |a_{ii}| \right)^2 \leqslant n\|\mathbf{A}\|_F^2$$

Taking the square root on both sides gives:

$$\sum_{i=1}^{n} |a_{ii}| \leqslant \sqrt{n}\|\mathbf{A}\|_F$$

Thus, we conclude:

$$|\mathrm{tr}(\mathbf{A})| = \left| \sum_{i=1}^{n} a_{ii} \right| \leqslant \sum_{i=1}^{n} |a_{ii}| \leqslant \sqrt{n}\|\mathbf{A}\|_F$$

This completes the proof:

$$|\mathrm{tr}(\mathbf{A})| \leqslant \|\mathbf{A}\|_F \cdot \sqrt{n}$$

## F    VISUAL COMPARISON FOR DIFFERENT METHODS

This section shows the visual comparisons of when our ParaSolver and ParaDiGMS are applied to speed up DDPM, DDIM, and DPMSolver on Stable Diffusion v2. The results are shown in Figure 2, Figure 3, and Figure 4. We can see that our ParaSolver significantly outperforms the competitors, with faster speed to generate a better image.

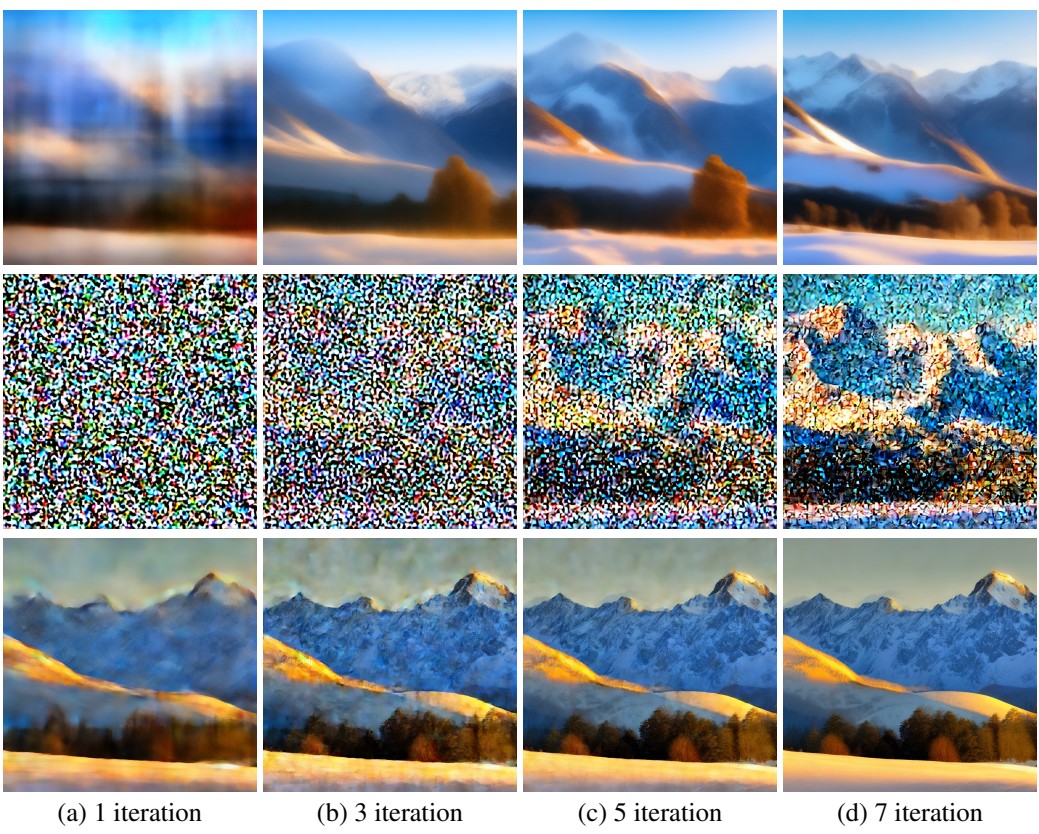

(a) 1 iteration          (b) 3 iteration          (c) 5 iteration          (d) 7 iteration

Figure 2: The intermediate generated images for expediting DDPM with 100 sequential steps on Stable Diffusion Model v2. The images in the first row are produced by DDPM. The images in the second row are created by ParaDiGMS. The images in the third row are generated by ParaSolver. For our ParaSolver, we set the number of subintervals as 100 and the preconditioning steps as 2.

## G    VISUAL COMPARISON FOR DIFFERENT SUBINTERVALS NUMBER $N$

This section presents visual comparisons illustrating the application of our ParaSolver to enhance the speed of DDIM and DPMSolver on Stable Diffusion v2 when using different $N$. The outcomes are depicted in Figure 5, Figure 6. It is clear that our ParaSolver performs better with smaller values of $N$, leading to enhanced output quality. For instance, in Figure 6, at $N = 20$, the image of the cute corgi at iteration 3 appears natural without any noise, whereas at $N = 100$, some noise is noticeable at this stage. This observation supports our claim that a system of nonlinear equations with $N = 100$ variables is more prone to accumulating errors during refinement compared to a system with only $N = 20$ variables.

## H    VISUAL COMPARISON FOR DIFFERENT PRECONDITIONING STEPS $M$

This portion delves into the impact of the preconditioning steps $M$ on the visual outcomes produced by our ParaSolver, which is utilized to accelerate DDIM with 100 sequential steps across the Stable

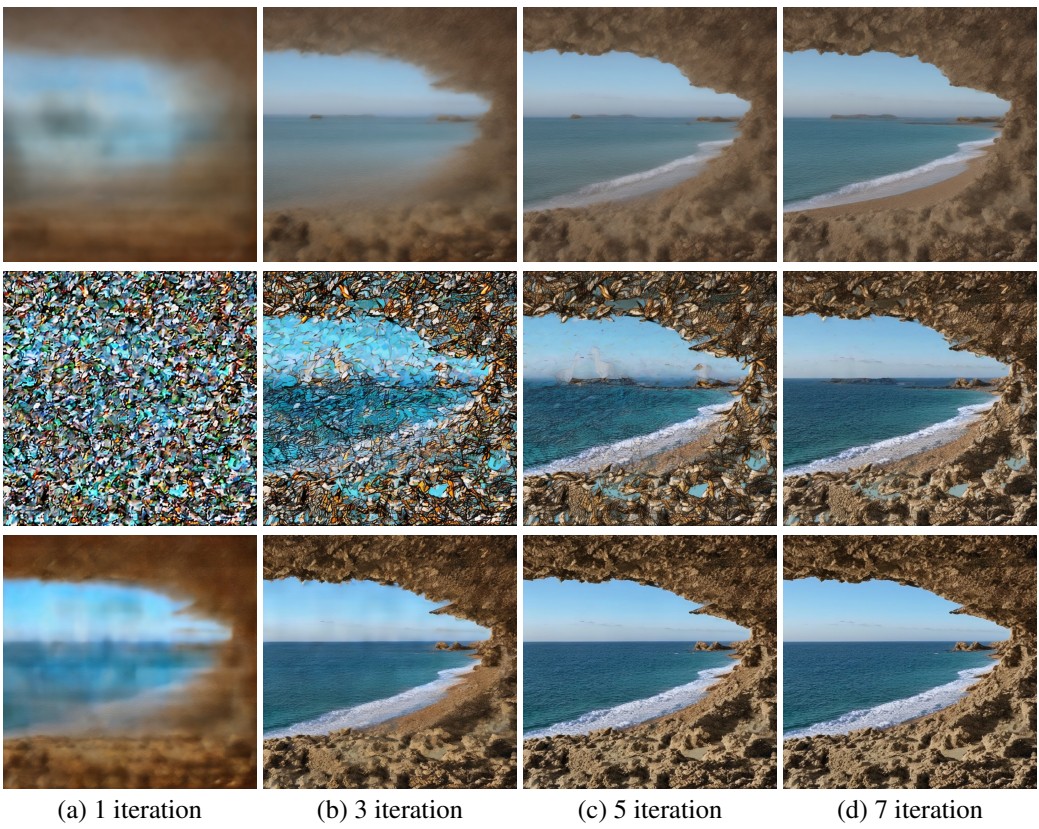

|     |     |     |     |
| :-: | :-: | :-: | :-: |
| (a) 1 iteration | (b) 3 iteration | (c) 5 iteration | (d) 7 iteration |

Figure 3: The intermediate generated images for expediting DDIM with 100 sequential steps on Stable Diffusion Model v2. The images in the first row are produced by DDIM. The images in the second row are created by ParaDiGMS. The images in the third row are generated by ParaSolver. For our ParaSolver, we set the number of subintervals as 40 and the preconditioning steps as 2.

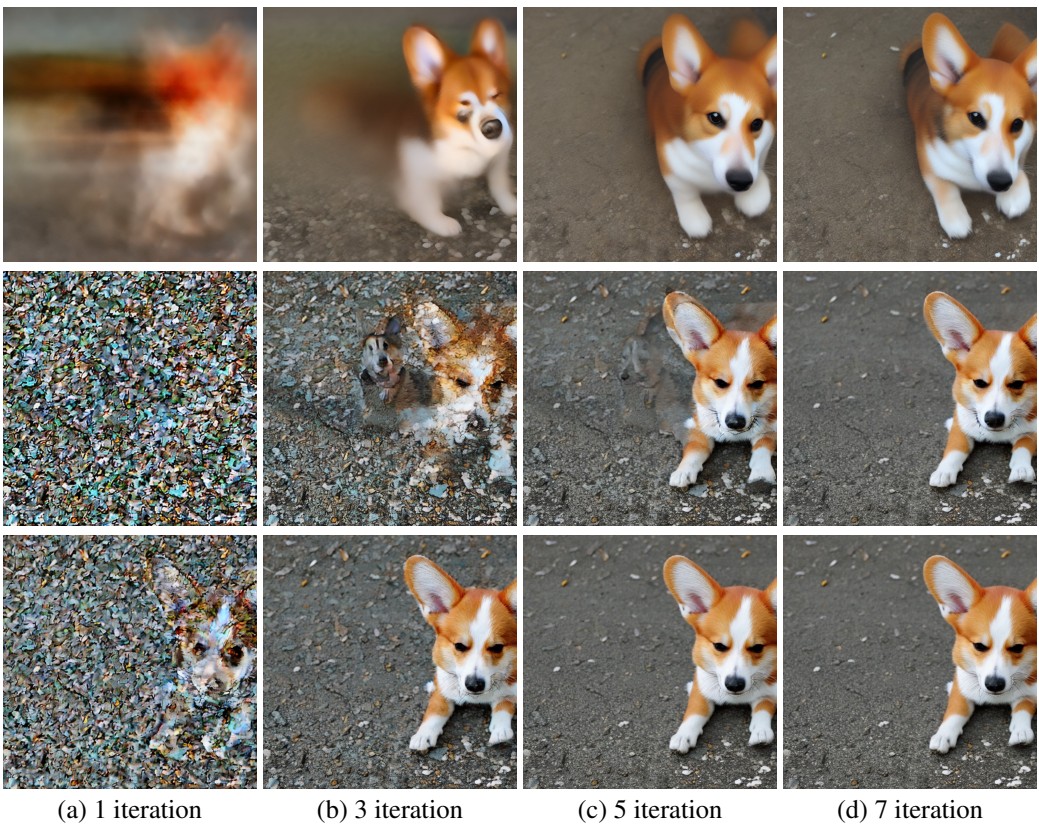

|                |                |                |                |
|:--------------:|:--------------:|:--------------:|:--------------:|
| (a) 1 iteration | (b) 3 iteration | (c) 5 iteration | (d) 7 iteration |

Figure 4: The intermediate generated images for expediting DPMSolver with 100 sequential steps on Stable Diffusion Model v2. The images in the first row are produced by DPMSolver. The images in the second row are created by ParaDiGMS. The images in the third row are generated by ParaSolver. For our ParaSolver, we set the number of subintervals as 40 and the preconditioning steps as 2.

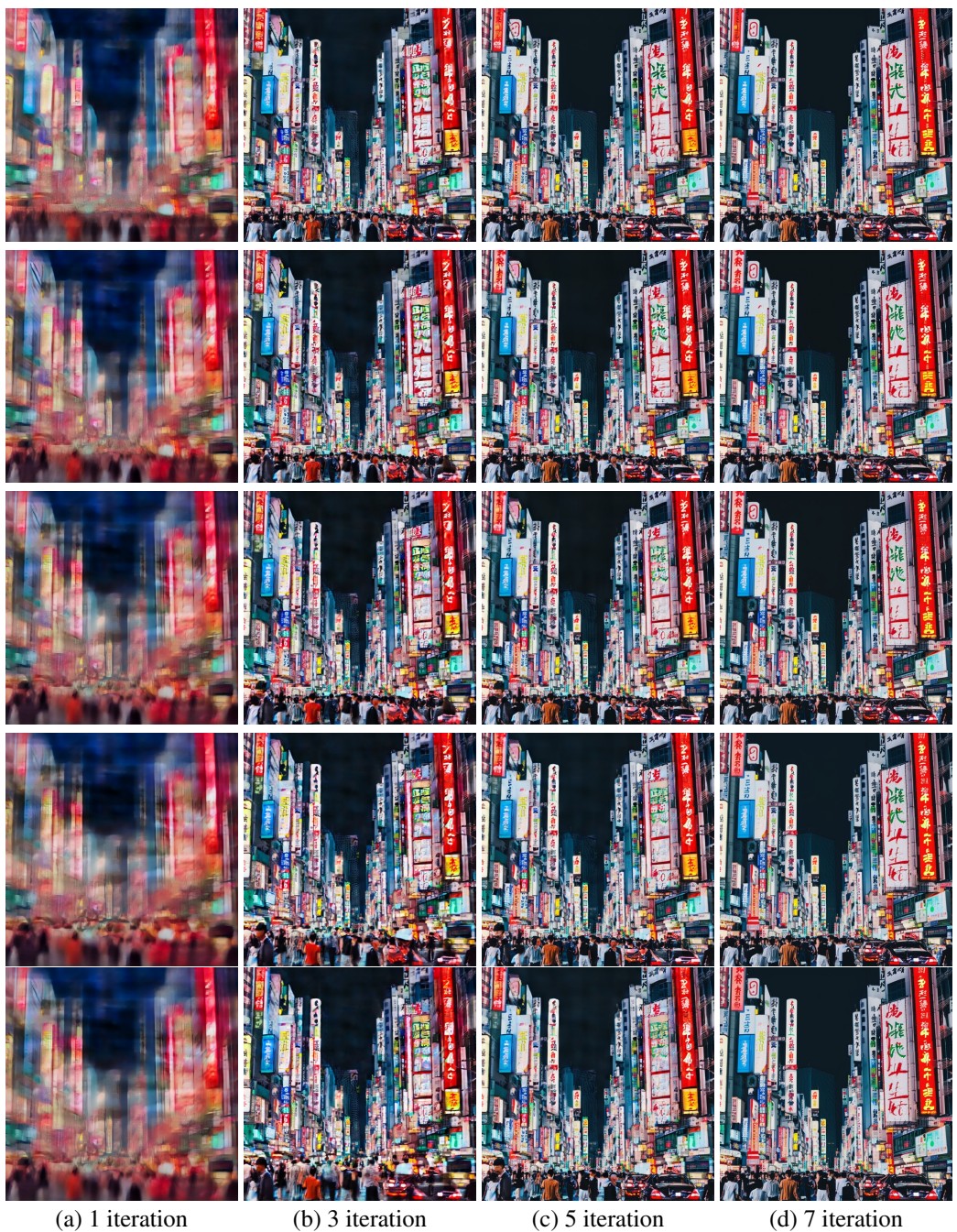

(a) 1 iteration  (b) 3 iteration  (c) 5 iteration  (d) 7 iteration

Figure 5: The effect of the number of the subintevals $N$ on the generated images when expediting DDIM with 100 sequential steps over Stable Diffusion Model v2. The images in the first row are produced by ParaSolver with $N = 20$. The images in the second row are created by ParaSolver with $N = 40$. The images in the third row are generated by ParaSolver with $N = 60$. The images in the fourth row are created by ParaSolver with $N = 80$. The images in the fifth row are generated by ParaSolver with $N = 100$. For our ParaSolver, we set the preconditioning steps as 2.

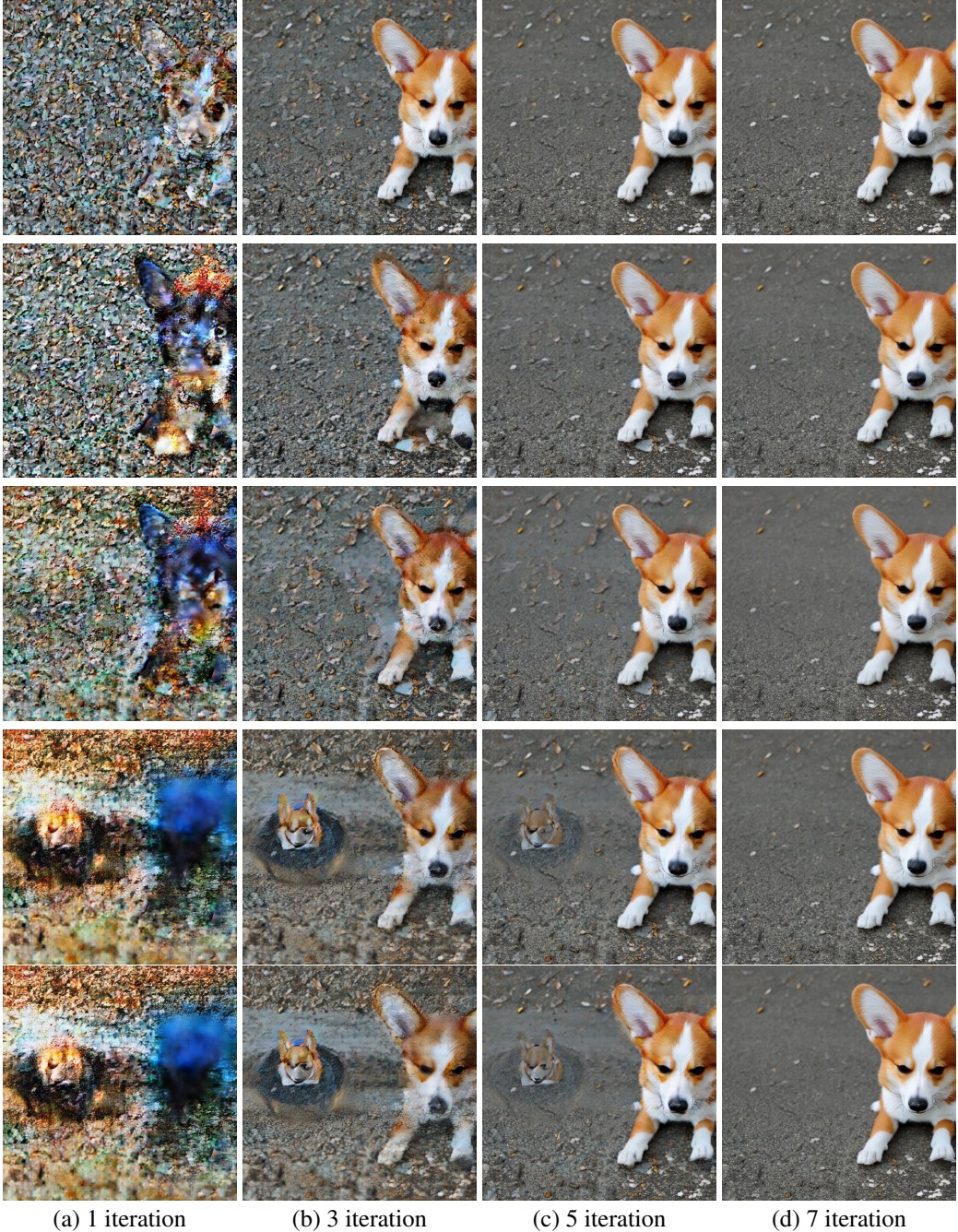

|                |                |                |                |
|:--------------:|:--------------:|:--------------:|:--------------:|
| (a) 1 iteration | (b) 3 iteration | (c) 5 iteration | (d) 7 iteration |

Figure 6: The effect of the number of the subintevals $N$ on the generated images when expediting DPMSolver with 100 sequential steps over Stable Diffusion Model v2. The images in the first row are produced by ParaSolver with $N = 20$. The images in the second row are created by ParaSolver with $N = 40$. The images in the third row are generated by ParaSolver with $N = 60$. The images in the fourth row are created by ParaSolver with $N = 80$. The images in the fifth row are generated by ParaSolver with $N = 100$. For our ParaSolver, we set the preconditioning steps as 5.

Diffusion Model. The outcomes are depicted in Figure 7. An intuitive observation emerges: elevating the number of preconditioning steps notably boosts the sample quality. Notably, even with just one parallel iteration, the resulting image appears impressive.

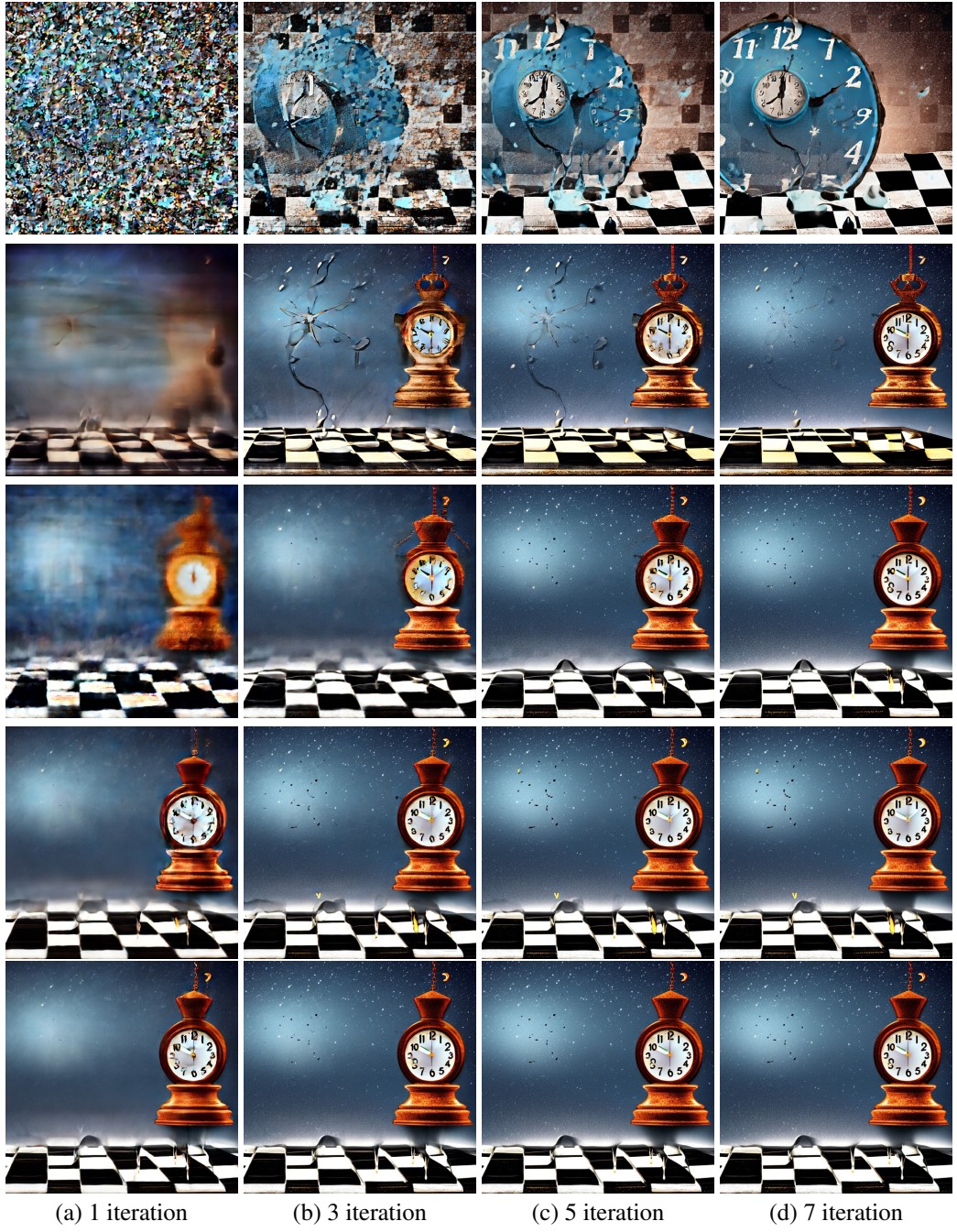

(a) 1 iteration  (b) 3 iteration  (c) 5 iteration  (d) 7 iteration

Figure 7: The effect of the preconditioning steps $M$ on the generated images when expediting DDIM with 100 sequential steps over Stable Diffusion Model v2. The images in the first row are produced by ParaSolver with $M = 0$. The images in the second row are created by ParaSolver with $M = 1$. The images in the third row are generated by ParaSolver with $M = 5$. The images in the fourth row are created by ParaSolver with $M = 10$. The images in the fifth row are generated by ParaSolver with $M = 15$. For our ParaSolver, we set $N = 40$.

