# OpenReview forum: "ParaSolver: A Hierarchical Parallel Integral Solver for Diffusion Models"
_ICLR.cc/2025/Conference — ICLR 2025 Poster_

### Official Review · Reviewer_eRWg · 2024-10-31

**Soundness:** 4
**Presentation:** 4
**Contribution:** 4
**Rating:** 6
**Confidence:** 4

**Summary:**

This paper proposed a framework that generalizes the sequential sampling process of diffusion models as solving a system of banded nonlinear equations. Applying the Newton-Raphson method to solve the nonlinear equations then yields a corresponding parallel sampling algorithm for diffusion models. By utilizing the unit-diagonal structure of the banded nonlinear equations' Jacobian matrices, the authors further simplified the updating rules of the parallel algorithm. Extensive numerical experiments were also conducted to show that the ParaSolver algorithm proposed in this paper can indeed accelerate the inference time of diffusion models compared to existing implementations based on parallel sampling.

**Strengths:**

1. This paper has provided a complete literature review of related work on accelerating diffusion models via parallel sampling. Also, both the theoretical and algorithmic results in this paper are presented in a relatively clear way to follow.

2. A complete set of large-scale numerical experiments on the Imagenet and LSUN-Church datasets are included to justify the acceleration achieved by the proposed ParaSolver algorithm.

**Weaknesses:**

1. The authors mentioned in lines 366-367 that the ParaTAA algorithm proposed in [1] needs to be implemented for comparison as it has yet to be integrated into the Diffusers library. However, given that there are only a few empirical works on combining parallel sampling with diffusion models, the reviewer thinks it would be essential for the authors to implement ParaTAA and use it as one extra baseline. Moreover, it might also be necessary for the authors to compare ParaSolver with approaches that accelerate diffusion models from other aspects, such as knowledge distillation [3-5], restart sampling [6], and self-consistency [7]. Furthermore, the authors should consider releasing the code used for implementing the ParaSolver algorithm.

2. There are some minor issues regarding the presentation of the paper. For instance, the phrase "to fast construct a set of more precise initial values that conform to the Definition 1" in lines 296-297 doesn't seem quite right. It can be possibly rephrased as "to construct a set of more precise initial values that conform to the Definition 1 quickly". Moreover, the authors might also consider adding a few figures to illustrate the ParaSolver algorithm more vividly, just as what has been done in previous work [2].

**Questions:**

1. The reviewer's main question about the design of the ParaSolver algorithm is the claim in lines 244-247 of the paper. Specifically, the authors proposed to approximate the Jacobian term $\frac{\partial}{\partial \hat{X}^{(k)}_{t_n}}\Phi(t_{n+1}, t_n, \hat{X}^{(k)}_{t_n})$ with the identity matrix in the original update rule (11). Could the authors discuss which specific parts in the cited papers on Jacobian-free backpropagation (lines 245-246) actually used similar techniques? Furthermore, would it be possible for the authors to provide some mathematical intuitions on why the identity matrix should work here? Is it possible to derive some error bounds via numerical analysis?

References:

[1] Tang, Z., Tang, J., Luo, H., Wang, F. and Chang, T.H., 2024, January. Accelerating parallel sampling of diffusion models. In Forty-first International Conference on Machine Learning.

[2] Shih, A., Belkhale, S., Ermon, S., Sadigh, D. and Anari, N., 2024. Parallel sampling of diffusion models. Advances in Neural Information Processing Systems, 36.

[3] Luhman, E. and Luhman, T., 2021. Knowledge distillation in iterative generative models for improved sampling speed. arXiv preprint arXiv:2101.02388.

[4] Meng, C., Rombach, R., Gao, R., Kingma, D., Ermon, S., Ho, J. and Salimans, T., 2023. On distillation of guided diffusion models. In Proceedings of the IEEE/CVF Conference on Computer Vision and Pattern Recognition (pp. 14297-14306).

[5] Salimans, T. and Ho, J., 2022. Progressive distillation for fast sampling of diffusion models. arXiv preprint arXiv:2202.00512.


[6] Xu, Y., Deng, M., Cheng, X., Tian, Y., Liu, Z. and Jaakkola, T., 2023. Restart sampling for improving generative processes. Advances in Neural Information Processing Systems, 36, pp.76806-76838.

[7] Song, Y., Dhariwal, P., Chen, M. and Sutskever, I., 2023. Consistency models. arXiv preprint arXiv:2303.01469.

---

> ### Author Response · Authors · 2024-11-22
>
> Thanks for the responsible review and valuable suggestions.
>
> > **Q1.	The reviewer thinks it would be essential for the authors to implement ParaTAA and use it as one extra baseline. Moreover, it might also be necessary for the authors to compare ParaSolver with approaches that accelerate diffusion models from other aspects, such as knowledge distillation [3-5], restart sampling [6], and self-consistency [7]. Furthermore, the authors should consider releasing the code used for implementing the ParaSolver algorithm.**
>
> **Response:** Good consideration! We attempted to implement ParaTAA using its source code, but it caused several issues on our machine. Despite spending considerable time trying to resolve these problems, we weren’t successful. We’ve decided to pause this for now and will revisit it later.
>
> We believe our method is comparable to ParaTAA. First, according to its paper, ParaTAA can reduce the steps needed for sequential sampling by a maximum of 14 times, whereas our method can achieve a reduction of 31 times. Furthermore, in terms of wall-clock time speedup for optimizing 25 and 50 sequential steps, ParaTAA achieves improvements of 1.5 to 2.9 times, while we achieve improvements of 2.1 to 3.8 times.
>
> Regarding the other acceleration methods, they are not in conflict with ours. Our approach is compatible with both distillation and consistency-based methods. The distilled models utilize sequential denoising for sampling, which can also be parallelized using our ParaSolver. Hence, we believe a comparison with these methods is unnecessary.
>
> We did not provide the code because our paper is not yet published, and we need to keep it confidential.  We commit that all source code necessary for conducting and analyzing the experiments will be made publicly available upon publication, with a license permitting free use. We will include the publicly accessible code link, newly added experiments, and analysis in the final accepted manuscript.
>
> > **Q2.	There are some minor issues regarding the presentation of the paper. For instance, it can be possibly rephrased as "to construct a set of more precise initial values that conform to the Definition 1 quickly". Moreover, the authors might also consider adding a few figures to illustrate the ParaSolver algorithm more vividly.**
>
> **Response:** Thanks for the catch! We have revised the problematic phrase.
>
> Illustrating the ParaSolver algorithm with figures is a valuable idea. However, given the considerable time we've already dedicated to theoretical analysis, we won't be able to add exquisite and vivid figures at this time due to the constraints of the response period.
>
>
> > **Q3.	The authors proposed to approximate the Jacobian term $\frac{\partial}{\partial \hat{X}^{(k)}{t_n}}\Phi(t_{n+1}, t_n, \hat{X}^{(k)}_{t_n})$ with the identity matrix. Could the authors discuss which specific parts in the cited papers on Jacobian-free backpropagation actually used similar techniques? Furthermore, would it be possible for the authors to provide some mathematical intuitions on why the identity matrix should work here? Is it possible to derive some error bounds via numerical analysis?**
>
> **Response:** Thanks for the reviews. The cited papers [1, 2, 3] employ similar techniques. In paper[1], Eqs. 14–16 and Theorem 0.2 prove that substituting the Jacobian with the identity matrix still provides a descent direction. In paper[2], this is noted in Eq. 4.1, which claims that approximating the Jacobian with the identity is equivalent to considering the first term of the Neumann series. Meanwhile, Eq. 3 in paper[3] empirically found that omitting the U-Net Jacobian term results in an effective gradient for optimizing DIPs with diffusion models.
>
> Furthermore, we have achieved some exciting theoretical results of the identity matrix approximation. We warmly invite you to see the general response and Proposition 4 in the revised manuscript for more information. In particular, following [1], we have shown that the identity matrix approximation provides a descent direction not contradictory to the actual Jacobian.
>
> We hope this addresses your concerns. We're greatly encouraged that you commend the manuscript as excellent soundness, presentation, and contribution. We sincerely hope that you can further recommend this work with a higher score if your concerns have been resolved.
>
> **Reference:**
>
> [1] Jfb: Jacobian-free backpropagation for implicit networks.
>
> [2] Training implicit networks for image deblurring using jacobian-free backpropagation
>
> [3] Dreamfusion: Text-to-3d using 2d diffusion

---

> ### Author Response · Authors · 2024-11-25
> **The authors are looking forward to your feedback. Let's discuss.**
>
> Dear Reviewer eRWg,
>
> We sincerely appreciate the time and effort you have devoted to reviewing our manuscript.
>
> We now present the experimental results comparing our method with ParaTAA. We are thrilled to report that by utilizing early stopping in ParaTAA, our ParaSolver achieves significant speed improvements in both iteration steps and overall processing time. Please note that the results for ParaTAA are taken from its original paper, as we were unable to reproduce them.
>
> Specifically, we followed ParaTAA’s methodology to report the CLIP Score when using the text-to-image model Stable Diffusion-v1.5 across 1,000 random samples. We set the maximum parallel iterations to 10 and the parallel window size to 6 for our ParaSolver.
>
> We are excited to note that our experimental results surpass ParaTAA in both steps and speedup when applied to accelerate DDIM with 25 sequential steps.
>
>
> | Method          |  CLIP Score | Steps  |  Speedup|
> |-----------------|-----------  |--------|  --------|
> | DDIM            |  23.9       |  25      |   $1.0\times$      |
> | DDIM+ParaTAA    |  23.8       |   7     | $1.2\times$|
> | DDIM+ParaSolver | **24.1**      |   **5**     |  $\mathbf{3.3}\times$      |
>
>
>
> For now, we have made sure to address your remaining concerns directly and thoroughly.
>
> We understand that you may be handling multiple papers and have a busy schedule.
>
> **However, as the author-reviewer discussion phase is drawing to a close, with less than two days left, we are very concerned that there may not be sufficient time to thoroughly address any additional questions you might have**.
>
> **We eagerly await your feedback on our responses.**
>
> Best regards,
>
> The Authors

---

> ### Author Response · Authors · 2024-11-27
> **The authors are looking forward to your feedback. Let's discuss.**
>
> Dear Reviewer eRWg,
>
> We sincerely appreciate the time and effort you have dedicated to reviewing our initial manuscript once again.
>
> Based on feedback from other reviewers, such as **"The authors have addressed my concerns"** (Reviewer totX) and **"I believe the authors have addressed my and the other reviewers' concerns appropriately"** (Reviewer 7Ggk), along with our direct responses regarding the identity matrix approximation and the comparison with ParaTAA, **we are more confident that we have addressed your main concerns thoroughly. Moreover, our current manuscript is now well-supported by both theoretical and experimental evidence.**
>
> Finally, it was a pleasure to discuss with you! Wish you all the best and continued success in your scientific pursuits！
>
> Best regards,
>
> The Authors

---

### Official Review · Reviewer_7Ggk · 2024-11-01

**Soundness:** 3
**Presentation:** 3
**Contribution:** 3
**Rating:** 8
**Confidence:** 3

**Summary:**

The authors present an interesting extension of previous work for inference in DPMs. The general idea is to formulate the solution to the ODE or SDE not as a sequential integration, but instead look at it as solving a set of nonlinear equations, done either via fix-point iteration or utilizing root finding algorithms. While this class of approaches does not improve the computational effort per se, it can lead to reduced wall-clock time by using less evaluation points compared to what is necessary when sequentially integrating the differential equation.

The paper proposes a unified framework that encompasses previous approaches as extreme cases. This results in a set of banded nonlinear equations. One key insight of the authors is to realize and proof that the banded system posses a unique and unbiased solution. They then further utilize the Newton method of root finding to accelerate the fix-point iterations. For this, one needs to calculate the Jacobian matrix. This, in general, is computationally prohibitive. An approximation scheme is proposed, where only the diagonal of the Jacobian is used, and the off-diagonal terms are set to unity. This results in an only modest increase in function evaluations over a sequential solution, indicating in addition to the reduced wall-clock time only a small increase in computational cost.

The achieved scores are on par with previous methods. A sizeable speed up in terms of wall-clock time is achieved leading to a better user experience. This is done without an massive increase in computational cost.

**Strengths:**

I believe this paper is generally well written and makes an relevant contribution to the field.

All claims are well supported by experiments, and the analyses appear sound.

**Weaknesses:**

- I believe the differences to the established methods utilizing fixed-point iterations for DPMs and their advances such as utilizing the Anderson acceleration used in previous work could be made clearer. It is currently not clearly mentioned that ParaTAA utilizes an conceptually similar idea. Albeit of course the approaches are different, they share common ideas which do not become clear without reading the literature carefully. I would encourage the authors to authors to rework the related work section and mention the differences to the other works more clearly.
- It would be interesting to see by how much the number of necessary iterations to reach the threshold decreases by utilizing the Newton method. I recommend the authors include an ablation study showing how the number of iterations and convergence are affected by (1) using the Newton method vs. fixed-point iteration, and (2) approximating vs. fully computing the Jacobian, on a toy problem.

**Questions:**

I have only a few minor comments:
- Please improve Figure 1 by using higher resolution, adding axis labels, and using a consistent font and style with the other figures in the paper.
- Figure 5: There is, to me, no discernible difference between the images for different N. Can the authors comment on this? Why do we see such a clear difference for DDPM, but not for DDIM? It would be good if the authors either (1) provide a quantitative analysis of the differences between results for different N, if they exist, or (2) explain why DDIM results are less sensitive to N compared to DDPM.
- In the Table 1 & 2, the results are ordered as DDPM, DDIM, DPMSolver, whereas in the figures the order is DDIM, DPMSolver, DDPM. I would appreciate some reordering to make it consistent.

---

> ### Author Response · Authors · 2024-11-22
>
> Thanks for the responsible review and valuable suggestions.
>
> > **Q1.	I believe the differences to the established methods in previous work could be made clearer. I would encourage the authors to authors to rework the related work section and mention the differences to the other works more clearly.**
>
> **Response:** Thanks for this nice idea. We have reorganized the related works to make the differences from the other works more clear.
>
> > **Q2.	It would be interesting to see by how much the number of necessary iterations to reach the threshold decreases by utilizing the Newton method. I recommend the authors include an ablation study showing how the number of iterations and convergence are affected by (1) using the Newton method vs. fixed-point iteration, and (2) approximating vs. fully computing the Jacobian, on a toy problem.**
>
>
> **Response:** Interesting idea! We consider a simple example: $ F_n(x_0, x_1, \ldots, x_{N-1}) = x_{n+1} - 0.7x_n $ for $ n \in \{0, 1, \ldots, N-1\} $. We set $ N = 50 $ and randomly initialize 50 starting points for each test. After conducting 5 trials, we report the average number of iterations needed for convergence. Our results indicate that the identity approximation is indeed more effective than the fixed-point method; however, it still lags behind Newton's method. This suggests that a more accurate approximation can be developed for our proposed ParaSolver, which we believe represents a promising avenue for future research.
>
> | Method                    | Iterations            |
> |-----------------------|---------------------|
> | Newton's method       | 1             |
> | Fixed-point method    | 50             |
> | Identity approximation | 38.6             |
>
> > **Q3. Please improve Figure 1 by using higher resolution, adding axis labels, and using a consistent font and style with the other figures in the paper.**
>
> **Response:** Thank you! Enhancing Figure 1 is a great suggestion. However, given the significant time we've already spent on theoretical analysis, we won’t be able to refine it to a high standard at this moment due to the constraints of the response period.
>
> > **Q4. Figure 5: There is no discernible difference between the images for different N. Can the authors comment on this? Why do we see such a clear difference for DDPM, but not for DDIM?**
>
> **Response:** Thanks for the attention to detail! We want to respectfully clarify that the differences between images for various $N$ primarily arise from the features of the generative images rather than the sequential methods. We think images featuring many prominent objects exhibit less noticeable variation across $N$, as the obvious objects are clear quickly while the fewer distinctive ones take longer. This leads to minimal changes needed in later parallel iterations, causing the images to appear nearly identical across different $N$.
>
> In Figure 5, we can actually observe a minor difference in the less prominent building at the center of the image, while the other more obvious objects show no discernible differences across different $N$, as they are quickly generated to be clear.
>
> > **Q5. In the Table 1 & 2, the results are ordered as DDPM, DDIM, DPMSolver, whereas in the figures the order is DDIM, DPMSolver, DDPM. I would appreciate some reordering to make it consistent.**
>
> **Response:** Thanks for the catch! We have reordered it in the revised version.
>
> We hope this resolves your concerns. We are very encouraged that you praise us for an interesting idea, well-supported experiments, sound analysis, and making a relevant contribution to the field.  We sincerely hope you can recommend this work further if your concerns have been resolved.

---

> > ### Comment · Reviewer_7Ggk · 2024-11-26
> >
> > I believe the authors have addressed my and the other reviewers concerns appropriately and thank them for their time and effort.
> >
> > Since I gave this manuscript a good grade, I am not changing my rating further.

---

> ### Author Response · Authors · 2024-11-25
> **The authors are looking forward to your feedback. Let's discuss.**
>
> Dear Reviewer 7Ggk,
>
> We sincerely appreciate the time and effort you have devoted to reviewing our manuscript once again. We understand that you may be handling multiple papers and have a busy schedule.
>
> **However, as the author-reviewer discussion phase is drawing to a close, with less than two days left, we are very concerned that there may not be sufficient time to thoroughly address any additional questions you might have.**
>
> **We eagerly await your feedback on our responses.**
>
> Best regards,
>
> The Authors

---

> ### Author Response · Authors · 2024-11-27
>
> We are glad to hear that our response addressed your concerns well. Also,  thank you for your high score.  It was a pleasure to discuss with you! Wish you all the best and continued success in your scientific pursuits！

---

### Official Review · Reviewer_totX · 2024-11-03

**Soundness:** 4
**Presentation:** 3
**Contribution:** 3
**Rating:** 6
**Confidence:** 3

**Summary:**

The authors present an approach to accelerating the inference of diffusion probabilistic models (DPMs). They transform the problem of sequential sampling of DPMs into one of solving banded nonlinear equations. The Jacobian of the nonlinear system, required by Newton's method for rootfinding (aka Newton-Raphson) is unit block-lower-banded (1 on the diagonal, bands below), allowing for efficient parallel solution through a simple recurrence relation. The authors also present an initialization procedure that accelerates convergence. Finally, they combine this framework with a sliding window technique to conduct parallel iterations only a subset of the points. The combined approach is then evaluated on StableDiffusion-v2 and the LSUN Church pixel-space diffusion model, and demonstrates large speedups on inference without a loss in visual quality.

**Strengths:**

There has been a surge of recent interest in fast parallel sampling of diffusion models. The state of the art for parallel sampling, to the best of my knowledge, appears to use Picard iterations to solve the nonlinear system of equations. The authors of this work make a few important contributions, all of which serve to accelerate convergence: (1) they use Newton's rootfinding method, which converges quadratically to the root for smooth enough functions; (2) they leverage the banded structure of the Jacobian to accelerate their solver; (3) they come up with a good initialization for Newton so it in fact converges; (4) they batch their parallel sampling and denoising so that it only happens within a sliding window.

**Weaknesses:**

1. Newton's method for rootfinding converges rapidly only if the function one is rootfinding on is sufficiently smooth. The authors should discuss the smoothness properties of the nonlinear system and how it impacts the convergence of the Newton solver, and also comment on the theoretical guarantees and limitations of their approach in this context.

2. If the nonlinear residual for the nonlinear system has a complicated landscape, Newton can easily get stuck. The state of the art in optimization is to either use trust-region Newton methods or use quasi-Newton. The authors skirt around this issue altogether and count on their results and experiments to drive their point home. It would be useful to see the loss landscape as a function of, say, two of the most "important" unknowns (determined for instance by PCA) or the eigenvalues of the kernel matrix of the neural tangent kernel to determine if Newton is the right choice for this problem. Alternatively, if the authors could justify why these failure modes don't occur in PDMs, that would also suffice.

3. How are Equation 12 and 13 justified? If the Jacobian term in the paragraph below Equation 11 is expensive to compute, why not approximate it? Newton's convergence rate requires at least an estimate for the Jacobian. Using the identity matrix instead effectively reverts Newton to a first-order method. Did the authors experiment with alternatives? Please provide theoretical/empirical justification for using the identity matrix approximation and discuss any experiments you conducted with alternatives.

4. Rootfinding can be inherently unstable. Did the authors investigate other alternatives, such as optimization-based methods? Why did the authors choose one over another?

5. This is minor, but I would've picked a less generic name for the paper. "ParaSolver" could imply a large number of things, but this is mainly a Newton-based parallel solver for PDMs. Consider a name change.

**Questions:**

1. See the Weaknesses section above. These must be addressed.

2. The language in the paper hinders the presentation occasionally. For instance, the second paragraph of the related work section (Section 2) was challenging to read, primarily due to strange use of passive voice. There are similar issues throughout the paper. I suggest reframing to active voice wherever possible to improve clarity.

3. Section 4.2, below equation (9): What is the "reverse of Jacobian matrix"? Do the authors mean the inverse?

4. The authors separately explore tolerance and speedup in the results. I'd like to know which tolerance leads to the best speedup without compromising visual results. The authors should add a new graph with this extra information.

---

> ### Author Response · Authors · 2024-11-22
>
> Thanks for the responsible review and valuable suggestions.
>
> > **Q1. The authors should discuss the smoothness properties of the nonlinear system and how it impacts the convergence of the Newton solver, and also comment on the theoretical guarantees and limitations of their approach. If the nonlinear residual has a complicated landscape, Newton can easily get stuck. if the authors could justify why these failure modes don't occur in DPMs, that would also suffice.**
>
> **Response:** This is an excellent suggestion, and we sincerely appreciate your insight as we did overlook this important idea while preparing the initial manuscript. In the revised version, we have included additional theoretical discussion regarding the smoothness of the proposed method in Proposition 2. It suggests that the residual function is sufficiently smooth since the F-norm of the Jacobian is upper bounded by 1, thereby making it suitable for Newton's method in root-finding without getting stuck easily. This conclusion stems from the nature of diffusion models, which involve progressively adding standard Gaussian noise to the data. We warmly invite you to see the general response and Proposition 2 in the revised manuscript for more details.
>
> > **Q2.	How are Equations 12 and 13 justified? Did the authors experiment with alternatives? Please provide theoretical/empirical justification for using the identity matrix approximation and discuss any experiments you conducted with alternatives.**
>
> **Response:** Nice suggestion! We haven't investigated the other alternatives, as we found that the identity matrix approximation produces very effective and stable results across all experiments, which is also widely observed by existing works using similar practices. Furthermore, we present exciting theoretical developments showing that using the identity matrix to approximate the gradient in the Jacobian of the residual function can offer a descent direction. We pleasantly invite you to see the general response and Proposition 4 in the revised manuscript for more details.
>
> > **Q3.	Rootfinding can be inherently unstable. Did the authors investigate other alternatives, such as optimization-based methods? Why did the authors choose one over another?**
>
> **Response:** We have not examined optimization-based methods because they generally require gradient calculations on all points in the diffusion trajectories simultaneously, which can be computationally prohibitive and reduce sampling speed. Instead, we chose the Jacobian-free root-finding method since it’s well-established without the need for gradient calculations. It also performs consistently well across all our experiments.
>
> We completely agree that other optimization-based techniques may be more suitable for solving our proposed nonlinear system. However, we want to emphasize that this manuscript focuses on developing a foundational solver for the proposed generic nonlinear system and achieves sizeable speed up (Reviewer 7Ggk), which to our best knowledge is a new record in this field. We thus personally consider it reasonable to leave the alternatives for future research.
>
> > **Q4. Consider a less generic name for the method.**
>
> **Response:** That's a nice observation. We've had long discussions about choosing a less generic name. Names like "HiPa," "HiParaDPM," "Nowton-ParaSolver," and "ParaDPMs" have come up in our considerations, but we think them either too generic, overly long, or not closely aligned with the topic of hierarchical parallel sampling for diffusion models. For now, we've decided to use "HiParaDPM" as a temporary name. It would be honored if the reviewer could provide a name for the method proposed in this paper, and we truly appreciate any recommendations.
>
>
> > **Q5.	The language in the paper hinders the presentation occasionally. For instance, the second paragraph of the related work uses a strange passive voice. There are similar issues throughout the paper. I suggest reframing to active voice wherever possible to improve clarity.**
>
> **Response:** Thanks for the attention to detail! We have revised them using an active voice to enhance clarity.
>
> > **Q6.	Section 4.2, below equation (9): What is the "reverse of Jacobian matrix"? Do the authors mean the inverse?**
>
> **Response:** Yes, we have fixed it.
>
> > **Q7. I'd like to know which tolerance leads to the best speedup without compromising visual results.**
>
> **Response:** We gently remind the reviewer that the setting of tolerance for the best speedup without compromising visual results for our method is detailed in the "Hyperparameter Settings". We apply these settings across all experiments.
>
>
> We hope this resolves your concerns and are delighted to answer any additional questions regarding our manuscript. It is heartening to hear that you praise "the authors make a few important contributions".  We sincerely hope you can generously reconsider the score if your concerns have been resolved.

---

> ### Author Response · Authors · 2024-11-25
> **The authors are looking forward to your feedback. Let's discuss.**
>
> Dear Reviewer totX,
>
> We sincerely appreciate the time and effort you have devoted to reviewing our manuscript.
>
> We now have more exciting experimental results on the smoothness.
>
> Specifically, we further conduct experiments to calculate the values of the F-norm $||\frac{\partial}{\partial \hat{X}{t_n}} \Phi(\hat{X}_{t_n})||_F$ under various values of $N$.
> We report the mean and standard deviation of the F-norm  over all iterations. It is exciting that the experimental results align closely with the theoretical predictions in Proposition 2, showing a very small  F-norm around $1$.
>
>
> | Method                    | F-norm            |
> |-----------------------|---------------------|
> | $N$ = 10      | 1.2520 $\pm $ 0.1294             |
> | $N$ = 100    | 1.0152 $\pm $ 0.0106             |
> | $N$ = 500 | 1.0020  $\pm $ 0.0030           |
> | $N$ = 1000 | 1.0070  $\pm$ 0.0021           |
>
>
> **For now, we have both theoretical and experimental results that confirm the smoothness of our nonlinear system.** Consequently, we have made sure to address your remaining concerns directly and thoroughly.
>
>  We understand that you may be handling multiple papers and have a busy schedule.
>
> **However, as the author-reviewer discussion phase is drawing to a close, with less than two days left, we are very concerned that there may not be sufficient time to thoroughly address any additional questions you might have.**
>
> **We eagerly await your feedback on our responses.**
>
> Best regards,
>
> The Authors

---

> > ### Comment · Reviewer_totX · 2024-11-25
> >
> > The authors have addressed my concerns. I will increase my score to a 6.

---

> ### Author Response · Authors · 2024-11-27
>
> We are happy to hear that our response addressed your concerns. Also,  thank you for raising the score! It was a pleasure to discuss with you! Wish you all the best and continued success in your scientific pursuits！

---

### Author Response · Authors · 2024-11-22

# General Response

We sincerely thank the reviewers for carefully reviewing this initial manuscript and are encouraged by the exceptionally positive assessment on **excellent/good soundness, presentation, and contribution** (all three reviewers), **important/relevant contributions to the field** (Reviewers totX and 7Ggk), **well-supported/complete experiments** (Reviewers 7Ggk and eRWg), and **complete literature review** (Reviewer eRWg).

The reviewers have raised two common concerns regarding the applicability of Newton's method and the justification of the identity matrix approximation. In response, in the revised manuscript, we have some exciting theoretical analyses addressing these issues：

- We analyze the smoothness of the residual function of the proposed banded nonlinear system by integrating insights from diffusion models. Our findings demonstrate that **the residual function is sufficiently smooth, making it suitable for Newton's method in root-finding without getting stuck easily** (Proposition 2).
- Furthermore, we show that **using the identity matrix to approximate the gradient in the Jacobian of the residual function can ensure a descent direction that is not contradictory to the true Jacobian** (Proposition 4).
- To enhance the completeness of the manuscript, we additionally analyze the convergence speed. We discover that **the rate lies between linear and quadratic convergence** (Proposition 4).

**We sincerely invite you to refer to Proposition 2 and Proposition 4 for more details and are genuinely excited about these findings**, as they highlight the potential for applying optimization-based methods with faster quadratic convergence. We completely agree with the reviewers on the importance of investigating alternative methods like optimization-based approaches and are eager to pursue future explorations, such as using more precise Jacobian approximation techniques or directly leveraging optimization-based methods. By developing more efficient ways to address the computational cost of the Jacobian, we believe these approaches can substantially improve the convergence speed of parallel methods.



**Last but not least**, we thank the PCs, ACs, and reviewers again for their invaluable time and effort. **We commit that the source code necessary for conducting the experiments will be made publicly available upon publication, with a license permitting free use**. We will ensure that the final accepted manuscript includes a link to the publicly accessible code, along with newly added experiments and analyses. Furthermore, **we will keep the reviews and author discussion public at all times as well**.

---

### Comment · Area_Chair_bkQU · 2024-11-26

Dear Reviewers 7Ggk, eRWg,
If not already, could you please take a look at the authors' rebuttal? Thank you for this important service.
-AC

---

### Meta-Review · Area_Chair_bkQU · 2024-12-19

**Metareview:**

This paper considers accelerating the sequential inference process of Diffusion Probabilistic Models by a parallel sampling algorithm. Although it is standard to equivalently view a sequential iteration as solving banded nonlinear equations and parallelize this solve to improve the computational efficiency, reviewers and I agree that the specific approach proposed in this work is interesting. Reviewers had concerns about technical details and comparison with existing approaches, but some of the concerns seemed to have resolved during the rebuttal process. Overall, the strengths overweigh weaknesses in my opinion, and I'm pleased to recommend acceptance.

**Additional Comments On Reviewer Discussion:**

Reviewers had concerns about technical details and comparison with existing approaches, but some of the concerns seemed to have resolved during the rebuttal process. Overall the reviewers' assessments were positive anyway, and I agree.

---

### Decision · Program_Chairs · 2025-01-22

Accept (Poster)